# The Berkeley Function Calling Leaderboard (BFCL): From Tool Use to Agentic Evaluation of Large Language Models

Shishir G. Patil [1]   Huanzhi Mao [1]   Fanjia Yan [1]   Charlie Cheng-Jie Ji [1]   Vishnu Suresh [1]
Ion Stoica [1]   Joseph E. Gonzalez [1]

## Abstract

Function calling, also called tool use, refers to an LLM's ability to invoke external functions, APIs, or user-defined tools—an essential capability for agentic LLM applications. Despite its prominence, there does not exist a standard benchmark to evaluate function calling due to two reasons – the challenging nature of evaluating when a function call is valid, and the challenge of acquiring diverse, real-world functions. We present the Berkeley Function Calling Leaderboard (BFCL), a comprehensive benchmark designed to evaluate function calling in a wide range of real-world settings. The BFCL benchmark evaluates serial and parallel function calls, across various programming languages, using a novel Abstract Syntax Tree (AST) evaluation method that can easily scale to thousands of functions. We construct the benchmark using a combination of expert-curated and user-contributed functions and associated prompts. Finally, BFCL benchmark evaluates the ability of models to abstain and reason in a stateful multistep agentic setting. Evaluating a wide range of models, we observe that while state-of-the-art LLMs excel at single-turn calls, memory, dynamic decision-making, and long-horizon reasoning remain open challenges. Since its preview, BFCL has become the defacto standard for evaluating function-calls, and can be accessed at https://gorilla.cs.berkeley.edu/leaderboard.html

## 1. Introduction

Large Language Models (LLMs) have made significant strides in diverse domains, including conversational AI, reasoning, and creative multimodal tasks. However, agents powering use-cases such as coding, and knowledge discovery (Huang et al., 2024; Yao et al., 2022) often require LLMs to interface with external (e.g., web-search) to either retrieve up-to-date information or to enact actions which have real-world consequences. To aid in this, previous work such as Gorilla (Patil et al., 2024) and Toolformer (Schick et al., 2023) introduced techniques to train LLMs to use tools, and since then there has been growing interest in enabling LLMs to leverage external tools, a capability commonly referred to as *function calling* or *tool use*.

Despite substantial progress in leveraging function calling in LLMs, evaluating function calling remains challenging. Existing solutions, such as Gorilla—APIBench (Patil et al., 2024), propose methods for assessing function calls but exhibit notable limitations: APIBench's evaluation metrics focus primarily on functional correctness, overlooking semantic errors or variations in API usage. Other benchmarks—ToolBench (Guo et al., 2024), ToolSandbox (Lu et al., 2024), and the Nexus Function Calling Leaderboard (Srinivasan et al., 2023)—fail to capture real-world use cases or patterns such as parallel invocations. We discuss some of these limitations in more detail in the related work section.

To address these challenges, we introduce the **Berkeley Function Calling Leaderboard (BFCL)**, a large-scale, multi-task, multi-turn benchmark that distinguishes function-calling capabilities among LLMs by evaluating their ability to invoke the correct function call. BFCL is composed of four parts: (1) 'single-turn' that tests single-turn function-calling scenarios, including parallel function invocations and multiple function candidates; (2) 'crowd-sourced' which consists of $2,251$ curated from more than $67,000$ community contributed real-life function-calling datapoints; (3) 'multi-turn' featuring eight curated API suites and 1000 queries, assessing sustained context management and dynamic decision-making; and finally (4) 'agentic' spanning applications (Vu et al., 2023), database queries (Gao et al., 2024), and chatbot context management (Packer et al., 2024). This comprehensive structure spans single-function tasks to complex, multi-step real-world use cases, enabling a thorough evaluation of an LLM's adaptability,

---

[1]University of California, Berkeley. Correspondence to: Shishir G. Patil <shishirpatil@berkeley.edu>.

*Proceedings of the 42nd International Conference on Machine Learning*, Vancouver, Canada. PMLR 267, 2025. Copyright 2025 by the author(s).

correctness, and effectiveness in function invocation.

Evaluating LLMs' function-invocation capabilities poses unique challenges because deterministic validation typically requires executing the corresponding functions which complicates large-scale evaluation. BFCL overcomes this by introducing a novel validation strategy that obviates the need for function execution. Drawing inspiration from programming language literature, we employ Abstract Syntax Tree (AST) sub-string matching as a proxy for actual function execution, thereby facilitating scalable evaluations. To validate this approach, we utilize a subset of our dataset to evaluate models using the earlier mentioned execution approach and observe a strong correlation between BFCL's execution and AST metrics.

As models increasingly incorporate function calling capability, our phased release lets us compare if BFCL's single-turn dataset has possibly leaked into the training data of the latest models. To investigate this, through CharNLL as a measure, we compare LLMs' familiarity with the single-turn dataset against that of the crowd-sourced dataset which was released six months apart.

In summary, this work makes the following contributions:

1. BFCL is a diverse dataset of 5,551 question-function-answer pairs across multiple programming languages, including Python, Java, JavaScript, REST APIs, and SQL. This diversity ensures a comprehensive assessment of LLMs' function-calling ability across a range of domains and use cases.

2. A novel application of Abstract Syntax Trees (AST) based sub-string matching to serve as a proxy for function execution and enabling scalable, deterministic validation of function calls.

3. The first inclusion of community-contributed, real-world user queries and functions in function-calling evaluations, providing a more accurate representation of practical complexities and use cases.

## 2. Related Work

**Language Models Using Tools.** Function calling (Schick et al., 2023) extends the capabilities of LLMs beyond its own knowledge base by enabling them to interact with external tools and APIs. Unlike structured output (Zhong & Chen, 2021), function calling allows LLMs to perform tasks requiring real-time data or external computation (Attouche et al., 2024). Models like GPT-4 (OpenAI, 2024) have demonstrated early ability to generate structured JSON for function invocation, prompting research into leveraging API calls as functions. More recent models have natively integrated function calling, empowering them to interact

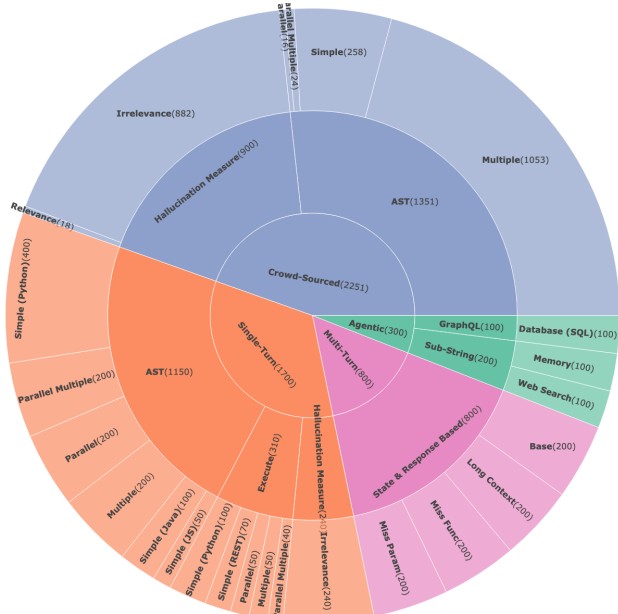

Figure 1: This chart visualizes the diverse categories within BFCL. The inner ring represents the four major sections, the middle ring specifies their respective evaluation methods, and the outer ring highlights each category. Numbers indicate the total number of datapoints in each category.

with external systems for knowledge retrieval (Sasaki et al., 2024) and real-time interaction. Furthermore, advancements like the LLMCompiler (Kim et al., 2024) optimize this process through parallel function execution, improving both efficiency and accuracy. This integration of function calling represents a crucial step towards more capable and versatile LLMs and subsequently unlocks broad agentic behaviors.

**Benchmark for Function Calling.** Quite a few benchmarks have been proposed to test LLM's ability to perform function calling, in this work, we focus on those designed to evaluate a model's native function-calling capabilities, rather than methods that rely on prompt-based or code-generation approaches (e.g., Nexus Raven (team, 2023) uses a prompt-based protocol, and AppWorld (Trivedi et al., 2024) emphasis code generation capability). Among benchmarks that do evaluate native function calling, many such as App Blend (Basu et al., 2024) and API Bench (Patil et al., 2024) focus solely on single-turn interactions. Although TinyAgent (Erdogan et al., 2024) addresses nested function calls, it does so by using placeholder variables instead of letting the model see the actual execution output of earlier calls; in effect, it still operates under a single-turn framework.

For benchmarks that truly cover multi-turn interactions, most are constrained by a narrow domain scope or a limited set of functions. TauBench (Yao et al., 2024), for example, supports only 28 functions spanning two domains (airline and retail), and RestBench (Song et al., 2023) offers scenarios solely within the TMDB and Spotify domains. The

narrow coverage ($< 150$ entries) is more prone to overfitting and does not sufficiently reflect the breadth of real-world function-calling scenarios. Furthermore, benchmarks like ToolSandBox (Lu et al., 2024) and TauBench (Yao et al., 2024) rely on LLMs to simulate user queries. Despite careful attempt to control the user simulator's behavior, LLM-based users remain prone to hallucination and instruction-following errors, which confound evaluation.

Other works, such as ToolBench (Qin et al., 2023), depend solely on Rapid APIs that are subject to high variance in performance, making reproducibility a challenge. While the subsequent StableToolBench (Guo et al., 2025) version mitigates this by caching or simulating API responses, it continues to rely on LLM-based evaluators for determining response solvability, thus risking model-induced biases and undermining objectivity (e.g., GPT-family models tending to favor responses from their own model (Panickssery et al., 2024)). Similar issues exist in T-Eval (Chen et al., 2024), which also depends on LLM-based evaluation.

Lastly, current benchmarks uniformly employ LLM-curated user queries, limiting their ability to accurately reflect genuine user interactions.

Our benchmark directly addresses these shortcomings by incorporating deterministic evaluation metrics, an extensive and diverse set of robust multi-turn interactions, and a unique multilingual dataset derived from real-world, user-contributed queries that have more than 15 languages represented in user queries, including Chinese, French, Japanese, and Korean, etc. Additionally, on top of Python and REST (which are commonly covered in existing benchmarks), we also have entries in Java and JavaScript for more diverse programming languages.

Collectively, these enhancements enable a more comprehensive, fair, and reproducible assessment of LLM function calling, setting our benchmark apart from existing efforts.

**Impact:** Since it's preview, BFCL has become the de-facto evaluation for function calling used by all leading labs developing large language models (MetaAI, 2024; Team, 2025; Cohere, 2025). The leaderboard is constantly evolving and is in it's current iteration (v4). This paper collapses the timeline and condenses all the learnings

## 3. Berkeley Function Calling Leaderboard

BFCL employs a structured data curation pipeline to construct our benchmarking dataset across different categories. The pipeline follows five stages: **data collection** sources functions from online repositories, APIs, and user queries; **data pre-processing** extracts and structures key function attributes; **data generation** standardizes functions and user queries into a schema that can be presented to the

Figure 2: Examples of single-turn function-calling scenarios to illustrate multiple, parallel, and irrelevance entry types outlined in Section 3.1.

LLMs; **data transformation** augments data with incomplete queries, and **data validation** ensures consistency and correctness through comprehensive unit-tests.

### 3.1. Single-turn Dataset

We classify single turn function-calling scenarios into five types based on the number of available tools and their invocation patterns. **Simple** involves one tool with a single invocation, whereas **Multiple** includes several tools each invoked once. **Parallel** scenarios feature multiple invocations of a single tool, and **Parallel Multiple** combines multiple tools with multiple invocations. **Irrelevance** refers to cases where tools are available but not invoked. Detailed definitions are provided in Appendix C.

We evaluate function calling through two methods: AST-based substring matching and executable tests. For AST-based evaluation, we curate functions in Python, Java, and JavaScript from popular GitHub repositories, filtering out trivial ones. For executable tests, we include (1) Python functions covering mathematical and physical computations, and (2) API-wrapped functions emulating real-world services (e.g., currency exchange and geocoding). Functions are standardized into a schema to avoid inconsistent documentation and ensure fair model comparisons. We further increase complexity by generating multiple and parallel calls, adding distractor functions, and testing robustness with missing parameters. See Appendix E for details.

### 3.2. Crowd-sourced Dataset

The crowd-sourced dataset contains 64,517 real single-turn user queries collected between 2024-02-26 and 2024-04-01, representing genuine function-call interactions from users. We remove duplicates via ROUGE-L and embedding based similarity, and exclude queries from public sets to avoid contamination. Human experts minimally edit queries for clarity and adherence to our function schema, preserving their original semantics. See Appendix E.2 for details. This dataset has the same dataset structure and categories as Section 3.1.

## 3.3. Multi-turn Dataset

**Multi-turn** covers an entire conversation between the user and the assistant. Each conversation includes multiple **turns**—a new user message—and each turn may involve one or more **steps**—individual interactions between the assistant and a tool or environment.

Our multi-turn dataset evaluates the LLMs's ability to handle queries that evolve over multiple turns. It is divided into four categories: **Base** category covers the basic multi-turn user queries asking everyday tasks and providing all necessary information. **Missing Parameters** tests the model's ability to recognize when critical parameter information is missing from the user request and cannot be inferred from the system. **Missing Functions** tests the model to identify when no available function can fulfill the user request. **Long Context** challenges the model's ability to maintain accuracy in multi-turn long context queries or function call results. Detailed definitions are provided in Appendix D.

To construct the dataset, we developed a custom API codebase across diverse domains like vehicle control, ensuring full transparency of API state and design. **Data generation** involves task generation of multi-turn queries that describe real-world tasks, along with specified initial API state configurations and function documents. Human annotators label ground truth trajectories from the generated multi-turn queries. **Data validation** includes question completeness, initial state verification, and function call sequence alignment with human-labeled ground truth. Detailed data generation and validation processes are outlined in Appendix E.3.

## 3.4. Agentic Dataset

The Agentic dataset is divided into three categories. We describe them in detail below. They share similar data curation pipeline as the single-turn dataset in Section 3.1.

### 3.4.1. WEB SEARCH DATASET

For this category, the LLM has two tools: a DuckDuckGo search function that retrieves webpage titles, snippets, and URLs, and a fetch function that extracts full webpage content. To ensure fairness across models with varying knowledge cutoffs, questions focus on recent but stable information, such as the 2024 TIME Person of the Year, rather than constantly changing data like stock prices.

### 3.4.2. MEMORY DATASET

Our memory dataset spans **five distinct domains**, each chosen to reflect a practical, real-world use case for LLMs—*college advising, customer support, medical assistant*, etc. Within every domain, the model is first given a concise *setting prompt* (e.g., "You are an academic advisor helping a sophomore plan their coursework") and then

participates in a sequence of consecutive conversations that gradually reveal user-specific facts. After each domain-level dialogue block, we record a *memory snapshot*. Evaluation queries are issued with an *empty chat history* but access to the stored snapshot, probing whether the model can accurately retrieve, add, overwrite, or delete information that was mentioned minutes (short-term) or many turns (long-term) earlier. This design lets us measure how well the model maintains and updates memory across both topic boundaries and temporal gaps, mirroring real deployment scenarios where sustained, personalized assistance is critical.

### 3.4.3. SQL DATASET

Each question in the SQL category can be succinctly translated into an SQL query. Traditional text-to-SQL tasks typically involve prompting language models to generate a valid SQL query, then evaluating the result via exact string matching or by querying a predefined database. In BFCL, however, we adopt a more structured approach by supplying the LLM with a JSON-based schema that defines fundamental SQL operations (e.g., SELECT, INSERT, UPDATE, DELETE). Each operation includes detailed structured parameters needed for a complete SQL query (e.g., WHERE, LIMIT, JOIN). This schema allows for a deterministic translation of function calls into SQL queries and supports nested or more complex queries through the composition of multiple function calls.

## 4. Evaluation Methodology

BFCL employs tailored evaluation protocols for each dataset category. The **Single-Turn** category use both *AST-substring matching* (Section 4.1) and *execution-response matching* (Section 4.2), whereas the **Crowd-Sourced** category relies solely on the AST matcher. The **Multi-Turn** category combines a *state-based* and a *response-based* checker (Section 4.4), and the **Agentic** category is evaluated with a strict *exact-match* criterion (Section 4.5).

We later quantify the agreement among the AST substring-matching and execution-response matching metrics (Section 4.3) to validate the reliability of the AST approach, highlight differences in inference strategies for prompt-only versus function-calling models (Section 5.1), and, finally, leverage perplexity ratios on the single-turn and crowd-sourced settings to detect potential data contamination in the BFCL single-turn corpus.

## 4.1. AST Substring Matching

Evaluating function calls by direct execution can be challenging due to the limited availability of executable functions and the laborious process of manual implementation, which curtails the diversity of functions available for testing.

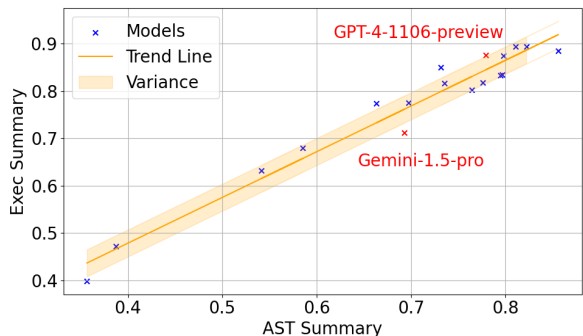

Figure 3: The Abstract Syntax Tree (AST) based evaluation (AST Summary) are strongly correlated with the evaluation by executing the functions (Exec Summary) validating AST as a reliable off-line evaluation methodology.

To address this, we introduce an Abstract Syntax Tree (AST) substring matching approach that preserves alignment with execution-based evaluation without actual execution.

We restrict the model's output to Python-callable function calls using prompt-based instructions, then extract function names and parameters through Python's `ast` module. Instead of requiring exact parameter matches, we verify that each parameter belongs to a predefined set of valid values. A function call is correct if the function name matches exactly and if all parameter values fall within their respective possible answers. For details on the AST matching rules, please refer to Appendix H.

### 4.2. Execution Response Matching

Execution response matching involves validating function calls by executing them and comparing the results against expected outcomes. There are three ways we compare the response. For functions that output deterministic results, we check for exact-match of the response. For functions whose outputs are time sensitive, we execute the ground truth function call and the model's output function call simultaneously and match the results, accounting for real-time value fluctuations. For nested lists or dictionaries, we perform structure matching, which only checks the length of the list and the presence of the dictionary key.

### 4.3. AST Matching Performance

By comparing the scores and relative rankings on the BFCL single-turn dataset, evaluated using AST and Execution in Figure 3, we observe a strong correlation between AST scores and execution-based performance. This suggests that AST matching serves as a reliable indicator of model effectiveness in real-world scenarios.

### 4.4. State & Response Based Evaluation

For multi-turn tasks, we employ two checks after each turn: **state-based** and **response-based**. An entry is correct only if it passes both checks in all turns.

State-Based Evaluation compares the system's final state after each turn (i.e., after all function calls) with the ground-truth state. Multiple sequences of function calls can achieve the same result, but the final state must match the labeled outcome. This approach captures modifications to the system (e.g., creating files or removing stocks from a watchlist).

Response-Based Evaluation verifies that the model follows the necessary sequence of function calls (the *minimal viable execution result path*) to produce the requested output. This is critical for read-only requests (e.g., retrieving stock prices), where we want to ensure the model calls the appropriate functions rather than guessing the result.

While state-based evaluation is a powerful technique, it cannot detect whether non-state-changing functions (e.g., `get_zipcode_by_city` or `estimate_distance`) were actually invoked. We need response-based checks to confirm the model is reasoning through the task reliably (e.g., calling `get_zipcode_by_city(City_Name)` before `get_weather_by_zipcode(City_Zipcode)`). By combining both types of evaluation, BFCL provides deeper insight into the model's correctness and decision-making process.

### 4.5. Exact-Match Evaluation

During evaluation, the model is given explicit formatting instructions through the system prompt detailed in Appendix J. We evaluate only the dedicated `answer` field with a strict exact-match criterion. Focusing on this individual field prevents spurious positives that would occur if the reference phrase appeared incidentally inside a longer, unclear sentence. For example, consider a yes/no question: a reply such as "I am not sure because no relevant information was found" contains the token "no," but the model has not actually committed to the negative answer. By isolating the `answer` field, such cases are not erroneously marked as correct responses.

Before matching, both candidate and reference answers are normalized—converted to lowercase and stripped of punctuation—using the same procedure as in our AST-based evaluation. A prediction is deemed correct if and only if the normalized strings are identical.

## 5. Results and Analysis

### 5.1. Accuracy

Table 1 presents the evaluation results of various LLMs on BFCL. While the top-performing models excel in single-turn, crowd-sourced, and hallucination-related metrics, there remains significant room for improvement in multi-turn and agentic tasks, particularly in memory management.

Table 1: Evaluating different LLMs on BFCL. The categories are defined in Section 4

| Model | Overall Acc | Single Turn AST | | | | Single Turn Execute | | | | Crowd Sourced AST | | | | Hallucination Measure | | Multi Turn | | | | Agentic | | |
|---|---|---|---|---|---|---|---|---|---|---|---|---|---|---|---|---|---|---|---|---|---|---|
| | | Simple | Multiple | Parallel | Parallel Multiple | Simple | Multiple | Parallel | Parallel Multiple | Simple | Multiple | Parallel | Parallel Multiple | Irrelevance | Relevance | Base | Miss Func | Miss Param | Long Context | Web Search | Memory | SQL |
| gpt-4o-2024-11-20 (Prompt) | 66.4 | 79.4 | 95.5 | 94.0 | 83.5 | 100.0 | 94.0 | 86.0 | 77.5 | 84.9 | 79.8 | 87.5 | 75.0 | 83.8 | 83.3 | 59.0 | 41.0 | 35.5 | 55.0 | 64.0 | 6.0 | 78.0 |
| gpt-4o-2024-11-20 (FC) | 65.8 | 77.2 | 93.5 | 93.0 | 86.0 | 88.3 | 92.0 | 94.0 | 82.5 | 81.4 | 78.8 | 87.5 | 75.0 | 83.1 | 83.3 | 62.5 | 6.0 | 37.5 | 58.0 | 82.0 | 0.0 | 81.0 |
| GPT-4-turbo-2024-04-09 (FC) | 60.9 | 70.4 | 91.0 | 90.0 | 87.5 | 87.4 | 90.0 | 86.0 | 77.5 | 83.7 | 78.6 | 81.2 | 70.8 | 83.8 | 72.2 | 54.0 | 13.5 | 35.5 | 49.5 | 66.0 | 4.0 | 50.0 |
| GPT-4o-mini-2024-07-18 (FC) | 60.6 | 74.8 | 92.0 | 90.0 | 84.0 | 83.3 | 92.0 | 84.0 | 75.0 | 78.7 | 76.2 | 87.5 | 70.8 | 74.7 | 81.8 | 47.5 | 19.5 | 29.0 | 40.5 | 78.0 | 6.0 | 66.0 |
| o1-2024-12-17 (Prompt) | 59.1 | 72.7 | 93.5 | 91.5 | 85.0 | 58.6 | 92.0 | 86.0 | 82.5 | 82.9 | 76.5 | 81.2 | 75.0 | 87.8 | 72.2 | 50.5 | 0.5 | 48.5 | 44.5 | 5.0 | 12.0 | 91.0 |
| Qwen2.5-72B-Instruct (Prompt) | 57.0 | 80.2 | 97.5 | 93.5 | 92.0 | 99.3 | 94.0 | 90.0 | 87.5 | 85.3 | 82.1 | 62.5 | 75.0 | 72.8 | 100.0 | 24.5 | 20.0 | 15.5 | 12.0 | 54.0 | 8.0 | 70.0 |
| Gemini-2.0-Flash-Exp (Prompt) | 56.8 | 76.8 | 95.5 | 95.0 | 92.5 | 63.6 | 92.0 | 84.0 | 80.0 | 85.7 | 79.3 | 81.2 | 87.5 | 86.4 | 77.8 | 28.0 | 3.0 | 19.0 | 21.5 | 73.0 | 0.0 | 53.0 |
| ToolACE-2-8B (FC) | 56.6 | 75.3 | 92.5 | 92.5 | 90.0 | 95.4 | 92.0 | 86.0 | 75.0 | 70.9 | 79.0 | 81.2 | 54.2 | 90.1 | 72.2 | 48.5 | 29.0 | 28.0 | 42.0 | 25.0 | 2.0 | 37.0 |
| Amazon-Nova-Pro-v1:0 (FC) | 56.6 | 68.8 | 92.5 | 92.0 | 84.5 | 97.1 | 84.0 | 84.0 | 77.5 | 80.2 | 77.5 | 81.2 | 58.3 | 71.0 | 77.8 | 37.5 | 19.0 | 22.0 | 26.0 | 76.0 | 0.0 | 51.0 |
| Qwen2.5-32B-Instruct (FC) | 55.6 | 72.8 | 94.0 | 93.5 | 88.5 | 97.6 | 88.0 | 84.0 | 77.5 | 80.2 | 80.1 | 43.8 | 62.5 | 81.9 | 64.7 | 29.5 | 25.5 | 20.5 | 13.5 | 53.0 | 4.0 | 45.0 |
| Gemini-2.0-Flash-Exp (FC) | 55.5 | 68.4 | 89.5 | 92.0 | 90.5 | 61.9 | 88.0 | 80.0 | 80.0 | 74.8 | 70.7 | 81.2 | 70.8 | 91.5 | 55.6 | 31.0 | 0.5 | 22.5 | 27.0 | 70.0 | 6.0 | 45.0 |
| BitAgent-8B | 55.0 | 76.2 | 95.0 | 94.0 | 82.5 | 98.6 | 94.0 | 88.0 | 77.5 | 77.9 | 77.4 | 87.5 | 70.8 | 82.4 | 83.3 | 48.0 | 40.0 | 26.5 | 39.5 | 22.0 | 0.0 | 29.0 |
| GPT-4o-mini-2024-07-18 (Prompt) | 54.5 | 80.1 | 90.5 | 89.5 | 87.0 | 62.9 | 96.0 | 82.0 | 82.5 | 81.4 | 76.7 | 93.8 | 79.2 | 80.7 | 83.3 | 33.0 | 12.0 | 17.0 | 26.0 | 43.0 | 0.0 | 63.0 |
| Claude-3.5-Sonnet-20241022 (FC) | 53.8 | 78.8 | 94.5 | 3.5 | 5.0 | 97.6 | 90.0 | 4.0 | 0.0 | 84.1 | 82.0 | 25.0 | 20.8 | 74.0 | 77.8 | 55.0 | 19.0 | 42.5 | 47.5 | 74.0 | 4.0 | 60.0 |
| o1-mini-2024-09-12 (Prompt) | 53.5 | 71.2 | 89.0 | 83.5 | 72.0 | 89.3 | 86.0 | 78.0 | 77.5 | 72.9 | 71.6 | 75.0 | 75.0 | 89.6 | 61.1 | 40.5 | 5.0 | 34.5 | 33.0 | 7.0 | 0.0 | 70.0 |
| o1-2024-12-17 (FC) | 52.8 | 67.9 | 93.0 | 0.0 | 0.0 | 60.6 | 94.0 | 0.0 | 0.0 | 81.8 | 79.0 | 0.0 | 0.0 | 82.6 | 72.2 | 52.5 | 38.0 | 30.5 | 43.0 | 48.0 | 12.0 | 83.0 |
| claude-3.5-haiku-20241022 (FC) | 52.3 | 68.0 | 92.0 | 2.5 | 0.0 | 87.9 | 90.0 | 24.0 | 0.0 | 78.3 | 78.8 | 18.8 | 0.0 | 63.7 | 83.3 | 54.5 | 26.5 | 35.0 | 44.0 | 83.0 | 6.0 | 59.0 |
| Qwen2.5-32B-Instruct (Prompt) | 52.1 | 70.2 | 94.5 | 90.5 | 88.0 | 96.6 | 90.0 | 90.0 | 82.5 | 83.0 | 78.5 | 62.5 | 58.3 | 73.8 | 100.0 | 25.0 | 20.0 | 15.0 | 11.0 | 46.0 | 0.0 | 42.0 |
| Amazon-Nova-Lite-v1:0 (FC) | 52.1 | 69.8 | 94.0 | 84.0 | 66.0 | 92.0 | 84.0 | 80.0 | 65.0 | 72.9 | 70.1 | 75.0 | 66.7 | 76.4 | 66.7 | 27.5 | 5.5 | 17.5 | 19.0 | 62.0 | 0.0 | 58.0 |
| GPT-4-turbo-2024-04-09 (Prompt) | 51.4 | 82.5 | 95.5 | 93.5 | 92.0 | 99.3 | 96.0 | 80.0 | 82.5 | 88.0 | 84.1 | 100.0 | 79.2 | 35.6 | 100.0 | 42.5 | 25.0 | 20.5 | 33.0 | 27.0 | 0.0 | 54.0 |
| Llama-3.1-70B-Instruct (Prompt) | 50.4 | 77.9 | 96.0 | 94.5 | 91.5 | 94.0 | 98.0 | 84.0 | 82.5 | 78.3 | 76.2 | 87.5 | 66.7 | 54.8 | 100.0 | 16.5 | 13.0 | 10.5 | 10.0 | 56.0 | 0.0 | 61.0 |
| Llama-3.3-70B-Instruct (Prompt) | 50.4 | 74.8 | 94.5 | 84.0 | 87.0 | 95.7 | 98.0 | 84.0 | 85.0 | 81.8 | 77.1 | 93.8 | 66.7 | 48.7 | 100.0 | 9.0 | 8.0 | 4.5 | 6.0 | 78.0 | 0.0 | 64.0 |
| Claude-3.5-Sonnet-20241022 (Prompt) | 49.4 | 81.4 | 92.0 | 70.5 | 46.0 | 100.0 | 92.0 | 72.0 | 85.0 | 86.3 | 80.1 | 81.2 | 45.8 | 64.4 | 77.8 | 9.0 | 5.5 | 5.0 | 10.5 | 68.0 | 0.0 | 60.0 |
| Qwen2.5-14B-Instruct (FC) | 49.2 | 69.7 | 95.0 | 88.0 | 89.0 | 90.4 | 92.0 | 72.0 | 85.0 | 77.1 | 75.0 | 75.0 | 70.8 | 77.7 | 55.6 | 19.5 | 17.0 | 16.5 | 10.5 | 30.0 | 0.0 | 30.0 |
| Hammer2.1-7b (FC) | 48.6 | 78.1 | 95.0 | 93.5 | 88.0 | 86.4 | 92.0 | 86.0 | 77.5 | 76.7 | 77.4 | 81.2 | 70.8 | 78.6 | 82.3 | 35.5 | 25.5 | 19.0 | 14.0 | 14.0 | 0.0 | 13.0 |
| Qwen2.5-14B-Instruct (Prompt) | 47.8 | 73.2 | 92.5 | 92.0 | 85.0 | 92.4 | 90.0 | 88.0 | 85.0 | 74.4 | 75.8 | 62.5 | 66.7 | 77.1 | 77.8 | 19.0 | 11.5 | 12.0 | 6.5 | 28.0 | 0.0 | 26.0 |
| claude-3.5-haiku-20241022 (Prompt) | 46.6 | 76.2 | 93.0 | 84.0 | 79.5 | 97.9 | 94.0 | 76.0 | 75.0 | 84.9 | 75.0 | 87.5 | 54.2 | 65.8 | 77.8 | 16.0 | 0.5 | 8.0 | 14.5 | 0.0 | 2.0 | 68.0 |
| Command-R-Plus (FC) | 46.5 | 72.1 | 89.5 | 82.5 | 64.0 | 90.9 | 90.0 | 84.0 | 60.0 | 70.5 | 58.8 | 62.5 | 45.8 | 53.2 | 72.2 | 16.5 | 10.0 | 9.0 | 17.0 | 69.0 | 0.0 | 45.0 |
| Command R7B (FC) | 46.4 | 68.2 | 91.5 | 85.5 | 81.5 | 87.1 | 92.0 | 82.0 | 75.0 | 63.2 | 58.7 | 56.2 | 62.5 | 81.0 | 55.6 | 6.5 | 1.5 | 6.5 | 5.5 | 69.0 | 0.0 | 18.0 |
| ToolACE-8B (FC) | 45.6 | 76.7 | 93.5 | 90.5 | 89.5 | 97.4 | 94.0 | 88.0 | 77.5 | 73.3 | 76.7 | 81.2 | 70.8 | 87.9 | 83.3 | 7.5 | 11.5 | 5.0 | 7.0 | 9.0 | 0.0 | 13.0 |
| Haha-7B | 45.2 | 78.1 | 95.5 | 89.5 | 81.0 | 80.4 | 96.0 | 88.0 | 80.0 | 78.3 | 77.6 | 75.0 | 70.8 | 80.7 | 83.3 | 13.0 | 10.0 | 11.5 | 7.0 | 20.0 | 0.0 | 8.0 |
| Hammer2.1-3b (FC) | 45.0 | 81.4 | 95.0 | 89.5 | 81.5 | 82.9 | 92.0 | 84.0 | 77.5 | 73.3 | 73.3 | 62.5 | 66.7 | 81.9 | 82.3 | 27.5 | 17.5 | 14.5 | 10.0 | 2.0 | 0.0 | 6.0 |
| Qwen2.5-7B-Instruct (FC) | 44.7 | 71.8 | 95.0 | 90.0 | 86.0 | 95.4 | 94.0 | 84.0 | 77.5 | 75.6 | 75.6 | 68.8 | 66.7 | 69.1 | 77.8 | 13.5 | 14.5 | 11.0 | 7.0 | 12.0 | 0.0 | 14.0 |
| xLAM-8x7b-r (FC) | 44.2 | 73.6 | 90.0 | 69.0 | 38.0 | 89.2 | 90.0 | 72.0 | 45.0 | 74.8 | 79.3 | 43.8 | 58.3 | 67.2 | 94.4 | 26.0 | 13.0 | 11.5 | 11.5 | 23.0 | 0.0 | 34.0 |
| Amazon-Nova-Lite-v1:0 (FC) | 43.7 | 69.8 | 94.0 | 84.0 | 66.0 | 92.0 | 84.0 | 80.0 | 65.0 | 72.9 | 70.1 | 75.0 | 66.7 | 76.4 | 66.7 | 27.5 | 5.5 | 17.5 | 19.0 | 10.0 | 0.0 | 9.0 |
| xLAM-7b-r (FC) | 43.0 | 74.2 | 95.5 | 81.0 | 73.5 | 74.0 | 96.0 | 82.0 | 67.5 | 72.1 | 74.9 | 50.0 | 62.5 | 77.1 | 94.4 | 16.5 | 8.5 | 7.5 | 7.5 | 14.0 | 0.0 | 9.0 |
| GoGoAgent | 42.9 | 75.4 | 93.0 | 92.0 | 84.5 | 95.4 | 96.0 | 88.0 | 80.0 | 72.9 | 75.4 | 68.8 | 66.7 | 83.1 | 77.8 | 1.5 | 2.0 | 0.5 | 0.0 | 10.0 | 0.0 | 14.0 |
| Qwen2.5-7B-Instruct (Prompt) | 42.7 | 75.3 | 94.5 | 91.5 | 84.5 | 92.1 | 90.0 | 86.0 | 85.0 | 76.7 | 74.9 | 62.5 | 70.8 | 65.2 | 88.9 | 9.5 | 8.5 | 7.0 | 5.5 | 14.0 | 0.0 | 15.0 |
| Llama-3.3-70B-Instruct (Prompt) | 42.4 | 74.8 | 94.5 | 84.0 | 87.0 | 95.7 | 98.0 | 84.0 | 85.0 | 81.8 | 77.1 | 93.8 | 66.7 | 48.7 | 100.0 | 9.0 | 8.0 | 4.5 | 6.0 | 36.0 | 2.0 | 8.0 |
| Gemma-3-12b-it (Prompt) | 41.4 | 77.3 | 95.0 | 90.0 | 73.0 | 84.7 | 94.0 | 88.0 | 72.5 | 84.9 | 70.8 | 87.5 | 62.5 | 61.1 | 88.9 | 8.0 | 3.5 | 2.5 | 4.5 | 25.0 | 0.0 | 13.0 |
| Hammer2.1-1.5b (FC) | 41.3 | 74.7 | 92.0 | 84.5 | 80.0 | 86.6 | 90.0 | 82.0 | 75.0 | 71.3 | 69.8 | 50.0 | 62.5 | 79.3 | 77.8 | 14.5 | 12.5 | 9.0 | 6.0 | 0.0 | 0.0 | 2.0 |
| Ministral-8B-Instruct-2410 (FC) | 40.9 | 71.8 | 91.5 | 84.5 | 87.5 | 71.3 | 86.6 | 86.0 | 77.5 | 72.3 | 62.5 | 56.2 | 45.8 | 55.3 | 70.6 | 21.5 | 8.5 | 10.0 | 5.5 | 13.0 | 0.0 | 12.0 |
| MiniCPM3-4B-FC (FC) | 39.9 | 69.8 | 91.5 | 82.5 | 79.5 | 89.3 | 90.0 | 86.0 | 85.0 | 74.8 | 63.9 | 43.8 | 62.5 | 72.2 | 72.2 | 5.0 | 1.0 | 3.0 | 1.5 | 6.0 | 0.0 | 8.0 |
| Amazon-Nova-Micro-v1:0 (FC) | 39.8 | 63.5 | 88.0 | 77.5 | 55.5 | 80.4 | 76.0 | 68.0 | 52.5 | 65.9 | 64.2 | 62.5 | 45.8 | 74.2 | 72.2 | 24.5 | 5.5 | 14.0 | 20.5 | 10.0 | 0.0 | 5.0 |
| Llama-3.1-8B-Instruct (Prompt) | 39.6 | 72.8 | 93.5 | 87.0 | 83.5 | 83.7 | 96.0 | 88.0 | 77.5 | 74.0 | 73.3 | 56.2 | 54.2 | 48.8 | 77.8 | 13.0 | 10.0 | 7.5 | 8.0 | 8.0 | 0.0 | 9.0 |
| Granite-20b-FunctionCalling (FC) | 39.4 | 72.8 | 91.5 | 84.0 | 81.5 | 84.9 | 92.0 | 86.0 | 82.5 | 68.2 | 56.3 | 43.8 | 58.3 | 74.8 | 88.9 | 6.0 | 1.5 | 4.5 | 1.5 | 0.0 | 0.0 | 29.0 |
| Qwen2.5-3B-Instruct (FC) | 38.7 | 73.3 | 92.0 | 73.5 | 76.5 | 86.9 | 90.0 | 86.0 | 70.0 | 74.0 | 72.1 | 62.5 | 45.8 | 64.3 | 88.9 | 8.5 | 6.0 | 4.5 | 5.0 | 3.0 | 0.0 | 5.0 |
| Falcon3-10B-Instruct (FC) | 37.1 | 70.5 | 93.5 | 87.5 | 87.0 | 97.1 | 92.0 | 92.0 | 82.5 | 76.4 | 76.2 | 50.0 | 41.7 | 31.9 | 94.4 | 6.0 | 5.0 | 4.5 | 4.5 | 12.0 | 0.0 | 8.0 |
| Qwen2.5-1.5B-Instruct (FC) | 36.6 | 72.4 | 87.0 | 81.5 | 75.5 | 88.0 | 90.0 | 78.0 | 72.5 | 74.0 | 66.1 | 50.0 | 45.8 | 62.7 | 94.4 | 4.0 | 1.5 | 3.0 | 1.5 | 0.0 | 0.0 | 6.0 |
| Qwen2.5-3B-Instruct (Prompt) | 35.7 | 74.2 | 90.5 | 79.5 | 79.0 | 80.9 | 86.0 | 80.0 | 80.0 | 69.8 | 66.5 | 56.2 | 62.5 | 54.2 | 88.9 | 5.5 | 3.5 | 2.0 | 2.5 | 0.0 | 0.0 | 4.0 |
| Llama-3.2-3B-Instruct (Prompt) | 35.6 | 73.8 | 92.0 | 80.5 | 76.0 | 87.3 | 92.0 | 78.0 | 77.5 | 64.0 | 64.9 | 12.5 | 45.8 | 51.7 | 88.9 | 8.5 | 2.5 | 4.5 | 5.5 | 0.0 | 0.0 | 5.0 |
| Falcon3-7B-Instruct (FC) | 35.4 | 64.8 | 89.5 | 86.5 | 88.5 | 89.0 | 94.0 | 86.0 | 77.5 | 74.0 | 66.5 | 75.0 | 62.5 | 33.7 | 88.9 | 3.5 | 3.5 | 3.5 | 3.0 | 8.0 | 0.0 | 6.0 |
| Qwen2.5-1.5B-Instruct (Prompt) | 35.3 | 71.0 | 86.0 | 70.0 | 66.5 | 80.4 | 94.0 | 88.0 | 80.0 | 70.5 | 59.3 | 56.2 | 41.7 | 63.0 | 83.3 | 1.5 | 2.5 | 0.5 | 0.0 | 0.0 | 0.0 | 3.0 |
| DBRX-Instruct (Prompt) | 35.0 | 73.5 | 92.0 | 42.5 | 37.0 | 90.1 | 88.0 | 46.0 | 52.5 | 78.3 | 73.0 | 75.0 | 41.7 | 40.5 | 94.4 | 0.0 | 0.0 | 0.0 | 0.0 | 9.0 | 6.0 | 36.0 |
| Hammer2.1-0.5b (FC) | 34.0 | 68.0 | 83.0 | 71.5 | 54.0 | 68.4 | 84.0 | 82.0 | 47.5 | 60.1 | 58.0 | 50.0 | 45.8 | 73.9 | 77.8 | 4.0 | 0.5 | 3.0 | 1.5 | 0.0 | 0.0 | 1.0 |
| Bielik-11B-v2.3-Instruct (Prompt) | 32.5 | 71.2 | 93.5 | 46.0 | 49.5 | 76.6 | 90.0 | 44.0 | 50.0 | 72.9 | 69.3 | 43.8 | 54.2 | 40.6 | 77.8 | 7.0 | 0.5 | 3.0 | 4.5 | 6.0 | 0.0 | 10.0 |
| GLM-4-9b-Chat (FC) | 30.1 | 65.2 | 81.5 | 0.0 | 0.0 | 94.0 | 90.0 | 0.0 | 0.0 | 72.5 | 64.4 | 0.0 | 0.0 | 79.7 | 66.7 | 3.5 | 4.0 | 2.5 | 4.0 | 3.0 | 0.0 | 8.0 |
| xLAM-7b-fc-r (FC) | 30.0 | 76.8 | 93.5 | 77.0 | 41.0 | 84.5 | 92.0 | 56.0 | 10.0 | 78.7 | 58.0 | 31.2 | 25.0 | 45.0 | 77.8 | 0.0 | 0.0 | 0.0 | 0.0 | 2.0 | 0.0 | 5.0 |
| MiniCPM3-4B (Prompt) | 29.2 | 63.5 | 72.5 | 65.5 | 62.0 | 40.4 | 34.0 | 48.0 | 80.0 | 46.5 | 34.8 | 43.8 | 41.7 | 74.4 | 50.0 | 3.0 | 3.5 | 1.0 | 0.5 | 0.0 | 0.0 | 3.0 |
| Gemma-3-4b-it (Prompt) | 28.9 | 64.3 | 91.5 | 56.5 | 41.0 | 68.1 | 80.0 | 30.0 | 12.5 | 72.9 | 62.8 | 37.5 | 29.2 | 48.1 | 77.8 | 0.0 | 0.0 | 0.5 | 0.0 | 0.0 | 0.0 | 6.0 |
| Meta-Llama-3-8B-Instruct (Prompt) | 27.3 | 62.7 | 82.5 | 48.0 | 50.0 | 77.7 | 86.0 | 42.0 | 60.0 | 61.2 | 61.4 | 37.5 | 33.3 | 82.6 | 77.8 | 1.5 | 0.0 | 1.0 | 0.5 | 2.0 | 0.0 | 7.0 |
| Qwen2.5-0.5B-Instruct (FC) | 27.1 | 61.2 | 78.0 | 60.0 | 50.0 | 51.2 | 88.0 | 52.0 | 52.5 | 56.2 | 41.3 | 56.2 | 20.8 | 46.2 | 88.9 | 1.0 | 2.0 | 1.0 | 1.0 | 0.0 | 0.0 | 1.0 |
| Falcon3-3B-Instruct (FC) | 22.7 | 58.0 | 69.0 | 61.0 | 25.0 | 54.6 | 46.0 | 20.0 | 10.0 | 55.4 | 56.3 | 31.2 | 37.5 | 34.5 | 77.8 | 0.5 | 0.5 | 0.0 | 1.0 | 0.0 | 0.0 | 3.0 |
| Qwen2.5-1.5B-Instruct (Prompt) | 22.0 | 51.2 | 79.0 | 46.5 | 40.5 | 46.6 | 76.0 | 52.0 | 35.0 | 48.8 | 40.3 | 12.5 | 25.0 | 21.2 | 94.4 | 0.5 | 1.0 | 0.0 | 0.5 | 0.0 | 0.0 | 0.0 |
| Qwen2.5-0.5B-Instruct (Prompt) | 21.0 | 58.2 | 68.0 | 53.5 | 33.0 | 63.1 | 70.0 | 62.0 | 52.5 | 53.9 | 34.8 | 56.2 | 16.7 | 16.4 | 94.4 | 0.0 | 0.0 | 0.0 | 0.0 | 0.0 | 0.0 | 0.0 |
| Llama-3.1-8B-Instruct (FC) | 21.0 | 55.8 | 54.0 | 48.5 | 34.5 | 58.7 | 58.0 | 54.0 | 30.0 | 51.9 | 49.0 | 37.5 | 41.7 | 4.9 | 94.4 | 5.0 | 7.5 | 5.0 | 4.0 | 4.0 | 0.0 | 4.0 |
| Llama-3.1-70B-Instruct (FC) | 20.8 | 49.2 | 24.5 | 12.5 | 15.0 | 53.0 | 36.0 | 30.0 | 7.5 | 52.3 | 52.6 | 31.2 | 25.0 | 44.8 | 100.0 | 7.0 | 4.0 | 4.5 | 4.0 | 5.0 | 0.0 | 0.0 |
| xLAM-1b-fc-r (FC) | 18.7 | 71.7 | 86.0 | 5.0 | 2.0 | 77.8 | 90.0 | 4.0 | 0.0 | 64.0 | 53.4 | 6.2 | 0.0 | 6.7 | 100.0 | 0.5 | 0.0 | 0.0 | 0.0 | 0.0 | 0.0 | 0.0 |
| Llama-3.2-1B-Instruct (Prompt) | 15.4 | 29.2 | 33.5 | 36.0 | 15.0 | 34.1 | 28.0 | 34.0 | 5.0 | 31.4 | 7.6 | 12.5 | 4.2 | 59.7 | 38.9 | 0.0 | 0.0 | 0.0 | 0.0 | 0.0 | 0.0 | 0.0 |
| Falcon3-1B-Instruct (FC) | 13.1 | 3.6 | 6.0 | 17.5 | 9.0 | 9.4 | 4.0 | 18.0 | 15.0 | 4.7 | 2.4 | 0.0 | 12.5 | 87.2 | 0.0 | 0.0 | 0.0 | 0.0 | 0.0 | 0.0 | 0.0 | 1.0 |
| Gemma-3-1b-it (Prompt) | 12.5 | 43.5 | 38.5 | 2.0 | 2.0 | 34.0 | 44.0 | 4.0 | 0.0 | 31.0 | 10.5 | 0.0 | 0.0 | 30.9 | 50.0 | 0.0 | 0.0 | 0.0 | 0.0 | 0.0 | 0.0 | 1.0 |

Models that support native function-calling (FC), such as GPT-4, Gemini-1.5-Pro, Claude-3.5-Sonnet, can run BFCL directly by supplying all function definitions in their `tools` input field. In contrast, most models lack built-in function-calling capabilities. For these models, we use a prompt-based workaround: we guide them to produce structured function calls through the system prompt (detailed in Appendix A), placing the function definitions in the system prompt rather than in a dedicated `tools` field. Throughout this paper, we refer to models that have their native function-calling feature enabled as "FC models" (or operating in "FC mode"), and those for which we rely on system prompts to trigger function calls as "prompting models" (or in "prompting mode").

If a model supports both FC and prompting modes, we find that the FC mode outputs tend to be structured responses that lower parsing errors. However, these structural constraints of the FC mode can limit the flexibility of a model in complex function calling scenarios. We therefore see more capable models often performing better in the prompting mode. For instance, Claude cannot execute parallel function calls in FC mode, whereas it can in prompting mode. In addition, when handling other programming languages (e.g., Java or JavaScript), models in prompting mode often outperforms FC mode.

Prompting models exhibit on average three times more decoding issues than FC models (412.93 vs. 182.5 out of 4,251 total entries), aligning with the observation that structured FC-mode outputs are easier to parse. However, among successfully decoded responses in the *multiple* function-call category, FC models show more incorrect function-call counts on average (77.5 vs. 21). A similar trend appears in the *parallel multiple* category, indicating that prompting models are generally more flexible in complex scenarios.

## 5.2. Dataset Composition Difference

We construct the data generation pipeline for single-turn based on our understanding of the composition of real-life function-calling scenarios from our experiences building a function-calling LLM, and from function-calling documentations (OpenAI, 2025b) and community forums (OpenAI, 2025a). At that time, there were no formal crowd-sourced

datasets. After we collected the crowd-sourced part of this dataset, we found that there are quite a few differences between the single turn and the crowd-sourced.

In crowd-sourced, there are significantly more scenarios on multiple and much less parallel function calling scenarios. This observation reflects how most users interact with function calling: high demand for the feature of having to intelligently choose between functions and lower demand for making parallel function calls in a single turn.

Crowd-sourced dataset also differ from single turn in that they contain multi-lingual user prompts, multi-lingual function docs, as well as user prompts that contains lots of redundant information, etc. As an example of the rich diversity we observed in our dataset, we even include a classification function triggered through function calling.

On average, each entry in crowd-sourced contains 3 function choices, with the maximum one having 37 function choices. Each function has an average of 4 parameters, with the maximum one having 28 parameters. Here are some statistics and distributions.

### 5.3. Parallel Function Call Ability

When a question requires multiple function calls (whether to the same function or different ones), the model can issue them all at once in a single turn, or sequentially across multiple turns. For tasks that have no interdependencies among calls, issuing parallel function calls in a single turn significantly reduces latency. For instance, checking the stock prices of 20 different stocks simultaneously is far more efficient than making 20 separate requests.

We observe notable shifts in the evolution of models' abilities to generate parallel function calls. Early models (e.g., the Claude Sonnet) lacked any function call capability. Later iterations introduced partial support for parallel calls—albeit with suboptimal performance. Interestingly, with the release of flagship versions such as `o1-2024-12-17-FC` and `claude-3-5-sonnet-20241022-FC`, the parallel function call feature appears to have regressed or even removed! We hypothesize that this is because although parallel calls can be more efficient for non-interdependent tasks, they may adversely affect accuracy when function calls are chained. In many real-world scenarios, each subsequent call relies on information returned by the previous one. As a result, generating calls one at a time—waiting for each execution's result—might ultimately be both faster and more accurate for these use cases.

### 5.4. Multi Turn Error Analysis

We analyze the errors made by models in two ways: The first is through a deterministic algorithmic approach for classifying the different ways a model's results don't align

with our ground truth answers. The second is an LLM-as-a-Judge approach to better understand the root-causes behind the the errors.

### 5.4.1. DETERMINISTIC ERROR ANALYSIS

We classify the errors shown by our benchmarked models into 5 categories: Empty Turn Response Error, Instance State Mismatch, Execution Response Mismatch, Force Termination, and API Error. An **Empty Response Error** refers to when a model does not make any function call for one or more turns in the conversation. An **Instance State Mismatch** refers to a discrepancy between the model's internal representation of the API state and the expected ground truth state. An **Execution Response Mismatch** error occurs when the responses generated by the model for a specific turn do not include all the expected responses defined in the ground truth for that turn. The **Force Termination** error occurs when the model's processing is abruptly stopped during inference. This happens in cases when the model uses many steps within a turn in an attempt to answer the user's question leading to termination of its efforts. Lastly, the **API Error** simply groups all the cases the model's provided API endpoint failed to run on an entry in the dataset due to it being down or the token length of our questions going over the maximum allowable input token length.

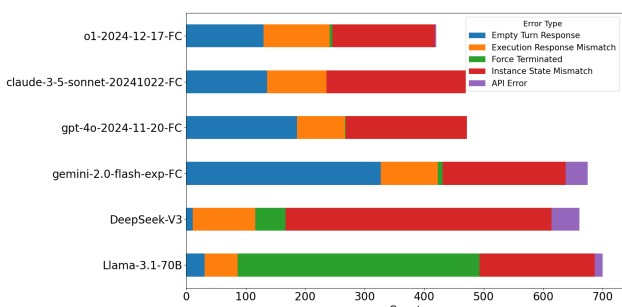

Figure 4: Error distribution across models on the BFCL Multi-Turn Dataset showing the counts of different error types: Empty Response Error, Execution Response Mismatch, Force Terminated, Instance State Mismatch, and API Error. The total number of entries in the dataset is 800.

### 5.4.2. LLM-AS-A-JUDGE ERROR ANALYSIS

In addition to the mechanical error analysis, we employed LLMs-as-judges to classify and analyze the root causes of failures in multi-turn interactions. We utilized a few-shot structured prompting method to query the LLMs about error cases. The prompts included detailed multi-turn conversation logs, initial configurations, and user queries. The judges were tasked with explaining and categorizing failures into predefined types: 1) **Failed to Understand Environment State**: Errors stemming from inaccurate assumptions or hallucinated environment states. 2) **Failed to Understand User's Request**: Misinterpretation of user request specifi-

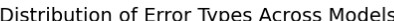

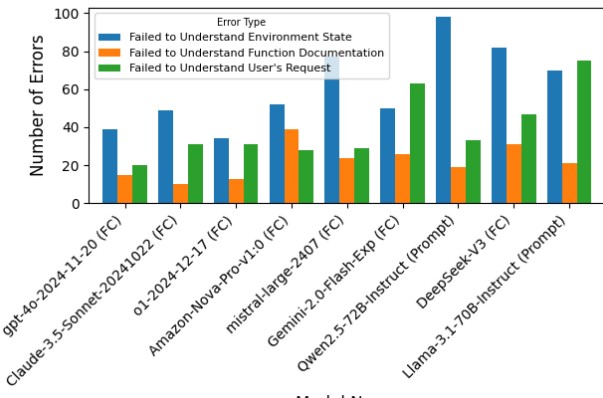

Figure 5: The bar chart shows the frequency of three error types—failed to understand environment state, failed to understand function documentation, and failed to understand the user's request—across various AI models. Error types are color-coded, illustrating differences in model performance.

cations. 3) **Failed to Understand Function Documentation**: Errors due to misinterpretation or misuse of provided function documentation. Full judge prompt template is in Appendix F

The most prevalent failure mode across models was **Failed to Understand Environment State** as can be seen in Figure 5. These errors occurred when the model either hallucinated or assumed incorrect environment state information, including attempting actions in incorrect directories or failing to execute necessary steps due to premature termination of actions due to state misalignment. **Failed to Understand User's Request** was the second most frequent error type. These failures typically arose when the model misinterpreted user intent, such as returning unsorted data instead of sorted content or failing to execute requested multi-step operations in the correct sequence user requested.

### 5.5. Measuring Data Contamination using crowd-sourced

Traditional benchmarks inadvertently compromise if a model's training data overlaps with the evaluation set, leading to artificially low perplexity and Negative log likelihood on those benchmarks. By comparing language modeling metrics between crowd-sourced and single-turn, we can diagnose possible data contamination or overfitting. In particular, an abnormally low perplexity or character-level negative log-likelihood (char-NLL) on the static single-turn benchmark, coupled with a significant performance drop on crowd-sourced, would signal that the model may have memorized the former. On the other hand, consistent performance across single-turn and crowd-sourced suggests genuine generalization rather than training exposure to the test answers

Table 2: While all models display consistently high perplexity in single-turn dataset then its crowd-sourced counterpart, the relative difference entails the model's familiarity to the single-turn which is open to be trained on as evaluation entries.

| Model | PPL (single-turn) | PPL (crowd-sourced) |
|---|---|---|
| Llama2-7B | 3.47 | 2.56 |
| Openfunctions-v2 | 3.09 | 2.49 |
| CodeLlama-7B | 3.45 | 2.49 |
| Meta-Llama-3 8B | 3.58 | 2.63 |
| Mistral-7B | 4.39 | 2.92 |
| Functionary-7B | 3.81 | 2.73 |
| Salesforce xLAM-7B | 3.67 | 5.09 |

Table 3: Character-level negative log-likelihood (Char-NLL) metric provides insights similar to those from Table 2. As the outputs of the function calls are structured, Char-NLL effectively captures model uncertainty and predictive performance at the token level across both single-turn and crowd-sourced settings. Lower values indicate better modeling of structured output.

| Model | Char-NLL (single-turn) | Char-NLL (crowd-sourced) |
|---|---|---|
| Llama2-7B | 0.344 | 0.264 |
| Openfunctions-v2 | 0.295 | 0.239 |
| CodeLlama-7B | 0.342 | 0.256 |
| Meta-Llama-3 8B | 0.322 | 0.245 |
| Mistral-7B | 0.404 | 0.295 |
| Functionary-7B | 0.337 | 0.254 |
| Salesforce xLAM-7B | 0.340 | 0.427 |

Tables 2 and 3 present the perplexity and char-NLL metrics, respectively, for several open-source models on the original single-turn benchmark versus the novel crowd-sourced dataset (wherein lower values signify superior predictive performance). It is observed that the majority of models achieve comparable or improved performance, i.e., lower perplexity and char-NLL on the crowd-sourced data relative to single-turn. For instance, Llama 2-7B (Touvron et al., 2023) exhibits a perplexity of 3.47 on single-turn compared to 2.56 on crowd-sourced, and a char-NLL of 0.344 versus 0.264. Similarly, Openfunctions-v2 and CodeLlama-7B also demonstrate slightly enhanced performance on the crowd-sourced queries. This trend implies that these models did not derive an unfair advantage from memorization on the original benchmark. This finding supports the inference that their strong benchmark scores are attributable to genuine capability rather than exposure to test solutions. In contrast, Salesforce xLAM-7B (Zhang et al., 2024) shows an increase in perplexity from 3.67 to 5.09(char-NLL from 0.340 to 0.427) on crowd-sourced, representing a notable performance degradation.

These performance decrements underscore the utility of the crowd-sourced dataset as a "stress test" for models: any model that has primarily memorized benchmark solutions or was overly tuned to the static test distribution is likely to be exposed through a significant score regression on the new dataset. In the instances of xLAM, the crowd-sourced evaluation reveals vulnerabilities that the original static test masked. It is noteworthy that both are specialized or smaller-

scale models, which may have been exposed to limited data variety; consequently, when confronted with genuinely novel queries, their performance markedly declines.

Overall, the perplexity and char-NLL metrics across single-turn and crowd-sourced present a coherent pattern. Figure 6, and 7 visualize these results, plotting each model's performance on single-turn against its performance on crowd-sourced. The majority of data points are situated near the diagonal, signifying comparable proficiency on both datasets. In contrast, the outlier models deviate sharply upward, indicating higher perplexity/NLL on crowd-sourced. This analysis substantiates the practical value of the BFCL Live methodology. By incorporating a real-world test set, a more robust indicator of contamination or overfitting is obtained. Models cannot rely on memorized answers for crowd-sourced queries; thus, any substantial discrepancy in metrics serves as an immediate flag for potential evaluation inflation. In summary, these comparative metrics confirm that top-performing models maintain strong generalization on fresh data, whereas models exhibiting any indication of contamination or poor robustness are clearly identified by the performance gap between their single-turn and crowd-sourced scores.

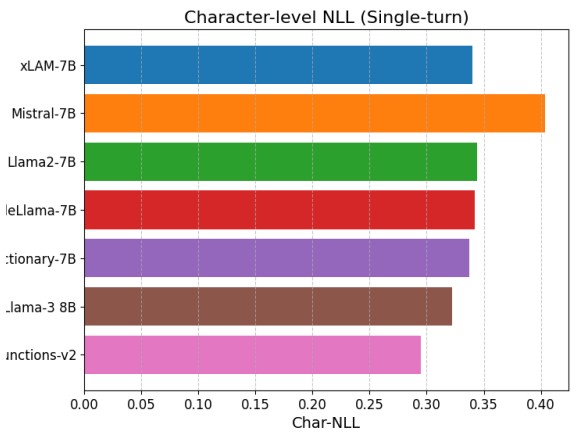

Figure 6: Character-level NLL on single-turn dataset.

### 5.6. Memory Management Category Analysis

Current models struggle with memory tasks; even the benchmark leader, *openai o1-2024-12-17 (FC)*, reaches only 12% accuracy. We observe models demonstrate faulty behaviors in memory management and organization. For instance, when preserving the fact that "the user is a fourth-year male CS major," some models save as one key (`user profile`), others split it (`major`, `year`, `gender`). Splitting quickly exhausts key space, demands frequent merges, and complicates retrieval.

Models often hallucinate during key retrieval. Because the key–value store requires exact matches, models should first call `list_keys` to view existing keys before retrieving

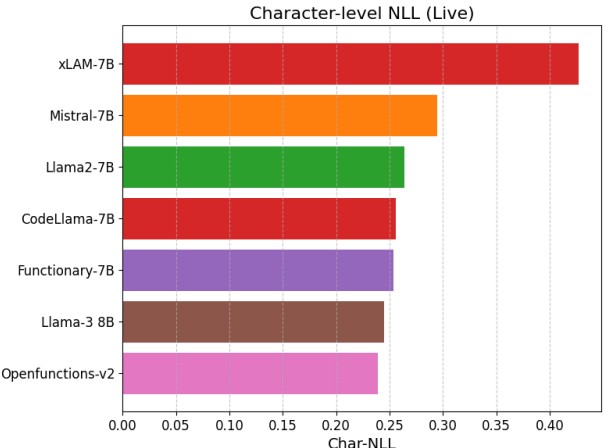

Figure 7: Character-level NLL on crowd-sourced dataset.

information. While some models handle this correctly, most skip listing the keys and instead attempt to guess a key when calling `retrieve`, leading to mismatches and errors. Even worse, many models give up after a single failure, mistakenly concluding that the information is unavailable rather than attempting to look up valid keys.

## 6. Conclusion

The Berkeley Function Calling Leaderboard (BFCL) establishes a new standard for evaluating large language models' (LLMs) ability to invoke and manage external tools and APIs. By introducing a comprehensive benchmark that spans single-turn, crowd-sourced, multi-turn, and agentic scenarios, BFCL provides a robust and scalable framework for assessing the function-calling capabilities critical to agentic AI systems. Its use of Abstract Syntax Tree (AST) based evaluation ensures reproducibility and avoids the scalability limitations of execution-based methods. The inclusion of real-world, multilingual user queries further enhances the benchmark's practical relevance.

Our analysis reveals that while many LLMs perform well on simple single-turn tasks, they often falter in more complex agentic and memory-intensive scenarios. This underscores the gap between current LLM performance and the demands of real-world, long-horizon reasoning tasks. Moreover, our comparative evaluations with crowd-sourced data expose potential overfitting in static benchmarks and highlight BFCL's role in stress-testing generalization.

As function-calling becomes a foundational capability for LLM-powered applications, BFCL serves as an essential tool for the community. We hope this benchmark not only drives advancements in LLM architectures and training but also promotes transparent and reproducible evaluation practices in the development of functionally robust AI agents.

## Impact Statement

The introduction of the Berkeley Function Calling Leaderboard (BFCL) represents a significant advancement in the evaluation and benchmarking of large language models (LLMs) for function-calling capabilities. Function calling is an increasingly critical skill for LLMs, enabling them to integrate seamlessly with external systems, perform complex tasks, and reason effectively in stateful, multi-turn interactions. Despite its importance, existing benchmarks inadequately capture the diversity, complexity, and real-world applicability of function-calling scenarios.

BFCL addresses these challenges by presenting a multi-faceted benchmark that evaluates LLMs across single-turn, multi-turn, crowd-sourced, and agentic datasets. By leveraging innovative techniques such as Abstract Syntax Tree (AST) substring matching for scalable and deterministic evaluation, and incorporating real-world user-contributed data, BFCL sets a new standard for evaluating LLMs' capabilities.

This benchmark has the potential to significantly shape the development of next-generation LLMs by providing researchers and practitioners with a comprehensive tool to assess and improve function-calling performance. Moreover, it paves the way for more robust, adaptable, and ethically-aligned LLM deployments in diverse domains such as healthcare, finance, and education.

Ultimately, BFCL contributes to the broader goal of making LLMs more effective, and reliable in real-world applications, fostering innovation and ensuring responsible AI use.

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

## A. System Prompt for Prompting Models

To enable prompting models to execute function-calling actions, we use the following universal system prompt, where the `{functions}` placeholder is replaced with the function documentation(s):

> """
>
> *You are an expert in composing functions. You are given a question and a set of possible functions. Based on the question, you will need to make one or more function/tool calls to achieve the purpose.*
> *If none of the functions can be used, point it out. If the given question lacks the parameters required by the function, also point it out.*
> *You should only return the function calls in your response.*
>
> *If you decide to invoke any of the function(s), you MUST put it in the format of [func_name1(params_name1=params_value1, params_name2=params_value2...), func_name2(params)]*
> *You SHOULD NOT include any other text in the response.*
>
> *At each turn, you should try your best to complete the tasks requested by the user within the current turn. Continue to output functions to call until you have fulfilled the user's request to the best of your ability. Once you have no more functions to call, the system will consider the current turn complete and proceed to the next turn or task.*
>
> *Here is a list of functions in JSON format that you can invoke.\n{functions}\n*
> """

## B. Data Augmentation on Function Documents

*Parallel Functions Category*: We created an additional user query that invoked the same function but with a different set of parameter values. This allowed us to expand the dataset by having multiple questions that invoked the same function, each with different input parameters. For example:

```
Parallel Functions Augmentation

query1 + [{'name': 'func1', 'description': 'order takeout'}] -> ans1
query2 (which contains query1) + [{'name': 'func1', 'description': 'order
takeout'}] -> [ans1, ans2]
```

The key transformation here is introducing `query2` and generating `ans2` based on a different set of parameter values.

*Multiple Functions Category*: In this category, we combined several function documents from different base entries to introduce distractor functions. However, the user query remained unmodified. These distractor functions were meant to test whether the model could accurately select the relevant function and not be misled by additional, unrelated function docs. GPT was used to ensure that none of the distractors could be alternative solutions to the function call. For example:

```
Multiple Functions Augmentation

query + [{'name': 'func1', 'description': 'order takeout'}] -> ans1
query + [{'name': 'func1', 'description': 'order takeout'}, {'name':
'func2', 'description': 'get weather'}] -> [ans1]
```

Here, the distractor `func2` is added to test the model's ability to focus on `func1` and avoid being distracted by irrelevant functions.

*Multiple Parallel Functions Category*: We combined multiple user queries and function documents from different base

entries, ensuring that more function docs were combined than user queries. This transformation tested the model's ability to handle multiple function calls and filter out unused functions. Some of the functions remained unused in the process, creating a more complex multi-query scenario. For example:

```
Multiple Parallel Functions Augmentation

query1 + [func1] -> ans1
query1 + query2 + [func1, func2, func3] -> [ans1, ans2]
```

The transformation here introduced `query2` and added `func2` and `func3` while testing how the model handles the multiple function calls.

*Function Parameter Removal*: Based on the base entry, we asked GPT to remove one or more pieces of parameter information from the user prompt, while keeping the function document unchanged. In this case, the model was expected to either ask a follow-up question to clarify the missing information or return an error message instead of making a function call (which would be considered a hallucination). This will be used in the irrelevance category. For example:

```
Function Parameter Removal Augmentation

query1 + [func1] -> ans1
query1' (missing parameter info) + [func1] -> [No Function Call, Model asks
for clarification.]
```

The key transformation involved removing the necessary parameter info in query1' and testing whether the model responded with a clarification or error message.

*Function Removal*: In this case, we removed one or more invoked functions from the function list in the augmented multiple parallel entries (from the previous step). The model was expected to either ask for more information on the missing function or produce an error indicating the absence of a relevant function for the query. This will be used in the irrelevance category. For example:

```
Function Removal Augmentation

query1 + query2 + [func1, func2, func3] -> [ans1, ans2]
query1 + query2 + [func1, func3] -> [No Function Call, Model asks for
clarification.]
```

The transformation involved removing `func2` from the list and verifying whether the model recognized its absence, producing the appropriate error message.

## C. Single Turn Dataset Formal Definition

### C.1. Dataset Structure

Each entry in the dataset consists of a user query, a list of candidate functions, and a corresponding expected model output:

- **AST-based functions**: Each entry is represented as:

$$(q, F, A), \text{ where } q \text{ is the user query, } F = \{f_1, f_2, \dots\} \text{ is the function set, and}$$
$$A \text{ is the labeled function and valid parameter set.} \tag{1}$$

- **Executable functions**: Each entry follows:

$$(q, F, \hat{f}(P)), \text{ where } \hat{f}(P) \text{ is the expected executable function call, and}$$
$$P = \{p_1, p_2 \dots\} \text{ is the set of parameters used in the function.} \tag{2}$$

### C.2. Dataset Categories

We categorize function-calling scenarios based on the number of available tools and the type of invocation:

- **Simple** ($\mathcal{S}$): Single available tool, single function invocation.

$$|F| = 1, \quad |\hat{f}(P)| = 1 \tag{3}$$

- **Multiple** ($\mathcal{M}$): Multiple available tools, single function invocation.

$$|F| > 1, \quad |\hat{f}(P)| = 1 \tag{4}$$

- **Parallel** ($\mathcal{P}$): Single available tool, multiple function invocations in a single turn.

$$|F| = 1, \quad |\hat{f}(P)| > 1 \tag{5}$$

- **Parallel Multiple** ($\mathcal{PM}$): Multiple available tools, multiple distinct functions called in parallel.

$$|F| > 1, \quad |\hat{f}(P)| > 1 \tag{6}$$

- **Relevance** ($\mathcal{R}$): At least one function call is expected.

$$|F| \geq 1, |\hat{f}(P)| \geq 1 \tag{7}$$

- **Irrelevance** ($\mathcal{I}$): Single or multiple available tools, but no function invocation is expected.

$$|F| \geq 1, \quad \hat{f}(P) = \emptyset \tag{8}$$

## D. Multi-turn Dataset Formal Definition

### D.1. Multi-turn Dataset Structure

**Dataset Structure**   Each dataset entry consists of a multi-turn user query sequence and a corresponding ground truth function trajectory:

$$(Q, T), \quad \begin{aligned} &\text{where } Q = \{q_1, q_2, ..., q_n\} \text{ represents the user queries,} \\ &T = \{\tau_1, \tau_2, ..., \tau_n\} \text{ represents the assistant's expected function call trajectory per turn.} \end{aligned} \tag{9}$$

### D.2. Dataset Categories

The dataset is divided into four categories:

- **Base** ($\mathcal{B}$): Standard multi-turn tasks where each turn has at least one expected function call.

$$\forall i, |\tau_i| \geq 1 \tag{10}$$

- **Missing Parameters** ($\mathcal{MP}$): One turn contains an underspecified query, and the expected assistant response is a natural language follow-up.

$$\exists i \text{ s.t. } \tau_i = \emptyset, \quad q_{i+1} \text{ resolves missing parameters.} \tag{11}$$

- **Missing Functions** ($\mathcal{MF}$): One turn requests a function that is unavailable, prompting the assistant to respond with a clarification request.

$$\exists i \text{ s.t. } \tau_i = \emptyset, \quad q_{i+1} \text{ provides missing function information.} \tag{12}$$

- **Long Context** ($\mathcal{LC}$): User queries and assistant responses in each turn contain a high number of tokens.

$$\forall i, \quad |q_i| + |\tau_i| \gg 1 \tag{13}$$

# E. Data Generation Pipeline Implementation Details

### E.1. Single-turn Dataset

**Data Collection**    For the single-turn tasks, we divide the data collection into two categories based on their evaluation method: **AST** categories that use Abstract Syntax Tree, and **Execute** categories that evaluate by execution. The evaluation methodology is discussed in 4.1 and 4.2.

*AST*: We collect functions from popular GitHub repositories (top 100 starred) in Python, Java, and JavaScript. These functions are well-documented, making them ideal candidates for our downstream tasks. We exclude trivial functions such as `__init__`, `__eq__`, and functions with fewer than two parameters (excluding the `self` parameter) to ensure complexity and relevance.

*Execute*: The category is divided into two sub-categories based on the backend type. 1) **Pure Python Functions**: We manually constructed functions inspired by common math and physics calculations. These are purely executable Python functions that don't rely on external APIs. 2) **Python Functions Wrapped APIs**: This sub-category includes functions that invoke API calls from popular public API providers such as ExchangeRate API, OMDb API, and Geocoding API. We focused on GET requests, as they are the most common in real-world scenarios. These functions demonstrate the model's ability to generate executable REST API calls through complex function documentation, using `requests.get()` along with the API's hardcoded URL and a description of the function's purpose and parameters.

**Data Preprocessing**    We pre-process them to extract useful context for downstream data generation tasks.

*AST*: We extract function names, descriptions, parameter names, types, and default values directly from signatures and docstrings.

*Execute*: For executable functions, we use Python's `requests.get()` as function document template. The schemas included base URLs, query parameters, path parameters, and body parameters.

**Data Generation**    We transform the extracted function information, such as docstrings from python functions and API documentation from ExchangeRate, into well-formatted function documents. This transformation ensures consistent formatting, including proper descriptions of parameters, types, and default values, making them compatible with our downstream evaluation pipeline. Once the function documentation was generated, realistic user questions were created based on these documents and their use in the original codebase or API context.

**Data Transformation**    To introduce complexity and mimic diverse real-world function-calling scenarios, we expand the dataset through various transformations detailed in B. These transformations included augmenting the entries to simulate different function calling patterns, such as parallel and multiple, and introducing scenarios with queries having incomplete or missing information to test the model's behavior.

**Data Validation**    We ensure 1) function documentations adhered to the BFCL format, including all required function schema fields. 2) The function parameters are precisely defined and correctly categorized. 3) User prompts were relevant, clear, and properly aligned with the corresponding function documentation. We've instructed three human experts

### E.2. Crowd-sourced Dataset

**Data Collection**    For the crowd-sourced dataset, 64,517 real-world user queries are collected between 2024-02-26 and 2024-04-01 via our hosted model endpoint.

**Data Preprocessing**    To preprocess the collected data:

*Deduplication*: We applied the ROUGE-L (Lin, 2004) score and OpenAI's text-embedding models to remove duplicate queries and function docs.

*Exclusion of Public Datasets*: We filtered out any queries from public test sets such as those from Nexus Function Calling Leaderboard to prevent contamination.

*Data Parsing*: The valid function documentation was then parsed into a JSON format compatible with the BFCL evaluation pipeline.

**Data Generation**    The expert-curated dataset doesn't have a data generation phase, because all entries are authenticate user-contributed data. The result from the pre-processing phase go directly into the transformation phase.

**Data Transformation**    *Minimum Edit Transformation*: Using the minimum edit principle, human annotators applied necessary corrections to improve clarity, precision, and consistency without changing the core content of the function docs or user prompts.

**Data Validation**    In addition to all the data validation step used in the single-turn section, we also make sure that 1) The transformed function doc and prompt preserve their original semantic meaning. 2) Any sensitive information in user prompts was replaced with placeholders to maintain privacy, and ambiguous content was clarified.

### E.3. Multi-turn Dataset

**Data Collection**    The multi-turn dataset began with the creation of a custom API codebase that spanned eight domains, including Vehicle Control, Trading Bots, Travel Booking, File System, Messaging, Twitter, Ticket Booking, and Math. Each API was designed to simulate real-world multi-turn function calls.

**Data Preprocessing**    We constructed a graph of function dependencies, where each function represented a node, and edges mapped output dependencies. This setup allowed us to model realistic multi-turn interactions across different APIs and domains.

**Data Generation**    The data generation process for multi-turn interactions involved:

- **Task Generation**: We prompt GPT-4o-0806a to invoke a series of function calls and then derive a natural language query that requires the function trajectories. The questions vary in tone and style to simulate different user personas and interaction scenarios.

  Precisely, we adopted the dataset from Persona Hub (Ge et al., 2024) to generate a diverse evaluation dataset with different personas ranging from people with different occupations, age groups, etc. For example, a persona can look like:

  High school physics teachers Science historians Elderly hermits Each persona would have a unique style to phrase the request.

- **Function Lists**: For each task, we provided a list of available functions from both primary and companion APIs.

- **Initial Configurations**: We set up initial states (e.g., pre-authenticated sessions) to avoid unnecessary interactions and focus on meaningful multi-turn tasks.

- **Human-labeled Ground Truth**: Expert human labelers reviewed and labeled each data point with ground truth for each multi-turn interaction.

**Data Transformation**    During data transformation, we scaled the dataset by sampling execution paths through the graph. Additionally, incomplete tasks were fixed by introducing additional configurations and function calls to maintain coherence.

**Data Validation**    Validation in multi-turn interactions involved:

- **Question Validation**: Ensuring that the questions were specific and complete.

- **Ground Truth Validation**: Verifying that the multi-turn function call sequences matched the ground truth.

- **Initial Configuration Validation**: Ensuring that the initial configurations were complete and relevant to the multi-turn tasks.

- **Function List Validation**: Checking that all necessary functions were included in the task's function list.

- **API Code Validation**: Using unit tests and format checkers to ensure that the API code was consistent and complied with the required standards.

## F. LLM Judge BFCL Error Analyzer Implementation Details

We use GPT-4o-08-06 as a judge, using bespoke curator (Marten et al., 2025) for structured synthetic data generation. The judge prompt is formatted as follows,

*„""*
*### **Error Analysis Prompt***

***Role:***
*You are an **error analysis expert** tasked with identifying and classifying*
*    failures in the AI assistant's responses. Your goal is to determine if and*
*    where the assistant failed, categorizing the root cause as one of the*
*    following:*

*– **Failed to Understand Function Documentation***
*– **Failed to Understand User's Request***
*– **Failed to Understand Environment State***
*– **No Failure***

***Instructions:***
*Carefully analyze the provided **multi−turn conversation** to identify any*
*    failures and their underlying causes.*

*−−−*

*### **Initial Configuration and User Queries***
*– **Initial Configuration:** {initial_config}*
*– **Related Function Documentations:***
*{function_documentation}*
*– **List of User Queries:** {user_queries}*

*−−−*

*### **Evaluation Process***
*Use the following guidelines to critically evaluate the multi−turn responses:*

*1. **Compare the model's function call traces** with the ground truth function*
*    call traces to identify any discrepancies in API usage.*
*2. **Compare the end state** with the ground truth state to determine if the*
*    model achieved the correct outcome.*
*3. **Pay attention to any mechanistic errors** reported by the state checker, as*
*    these indicate validation failures.*

*−−−*

*### **Entry ID: {entry_id}***
*#### **Turn−by−Turn Breakdown***

*{turns_breakdown}*

*−−−*

*### **State Checker Results***
*{state_checker_results}*

---

### **Failure Categorization**
A failure occurs **ONLY** if:
- The environment's **final state** differs from the **ground truth state**, OR
- The assistant **failed to produce the minimum required trajectory**.

**Exploration Steps**
- The assistant may take exploration steps, which may result in execution errors.
- **Do not penalize** the assistant for exploration steps.
- **Only mark** the critical step that leads to the failure.

If the **state checker reports an error**, a failure **definitely** occurred.
If the **state checker reports "None"**, check if the model failed to produce the **minimum required trajectory**.

---

### **Example Failure Analyses**
#### **Example 1**
**Context:** Fuel tank has 5 gallons of gas initially. Max capacity is 50 gallons.
**User:** *Fill the fuel tank until we are able to reach Rivermist. Oil costs money so I just need to reach there, I don't need a full tank.*
**Assistant Response:** `fillFuelTank(fuel_amount=50)`
**Ground Truth:** `displayCarStatus('fuel'), fillFuelTank(fuelAmount=44)`

**Failure Analysis:**
- **Failure Type:** *Failed to Understand Environment State*
- **Description:** The model filled the tank to maximum capacity without checking the current fuel level first.
- **Turn ID:** `0`

---

#### **Example 2**
**Context:** User has a trading account with $10,000 balance.
**User:** *I want to buy some Apple stock, but first tell me its current price and make sure I can afford at least 5 shares.*
**Assistant Response:** `execute_trade(symbol='AAPL', quantity=5, order_type='market', side='buy')`
**Ground Truth:** `get_stock_price(symbol='AAPL')`

**Failure Analysis:**
- **Failure Type:** *Failed to Understand User's Request*
- **Description:** The model executed a trade immediately when the user only requested price information and affordability check.
- **Turn ID:** `0`

---

### **Final Task**
Now, identify the failures using the above categorization. **Only identify ONE root cause of the failure.**

```
If there is **NO failure** in ANY turn, **report exactly ONE entry** with:
- **Failure Type:** 'No Failure'
- **Description:** 'No Failure Occurs'
- **Root Cause:** 'No Failure Occurs'
- **Turn ID:** '-1'

### **Your Failure Analysis:**
"""
```

## G. Function Calling Model Performance over time

PLease refer to Figure 7.

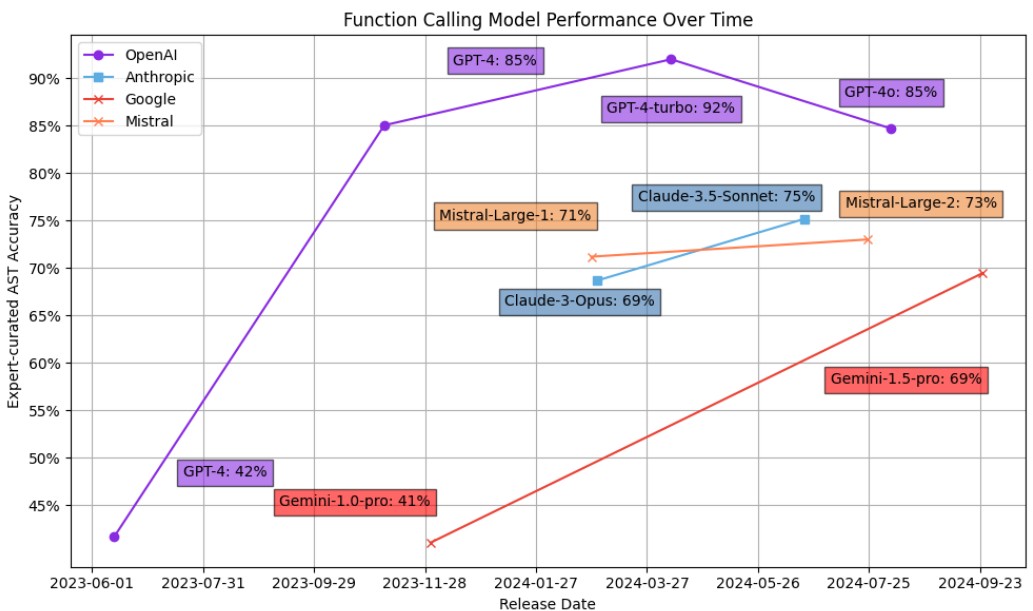

Figure 8: Models from 2023 and early 2024 struggled with reliable function calling at scale. As model sizes grew and function calling became a post-training objective, their capabilities improved significantly.

## H. BFCL AST Substring Matching

**Parallel Tool Calls**

By definition, *parallel* tool calls execute simultaneously; therefore, their order is irrelevant. If strict sequencing *is* required, the model must emit one function call at a time and wait for its completion before producing the next.

Given a predicted sequence $A = [a_1, a_2, \ldots, a_m]$ and a ground-truth sequence $B = [b_1, b_2, \ldots, b_n]$,

- we do **not** require positional alignment;

- any predicted call $a_i$ may match any ground-truth call $b_j$.

The evaluation follows an *all-or-nothing* rule: if even a single ground-truth call is unmatched, the entire prediction fails. This ensures the model identifies *all* required calls, regardless of order.

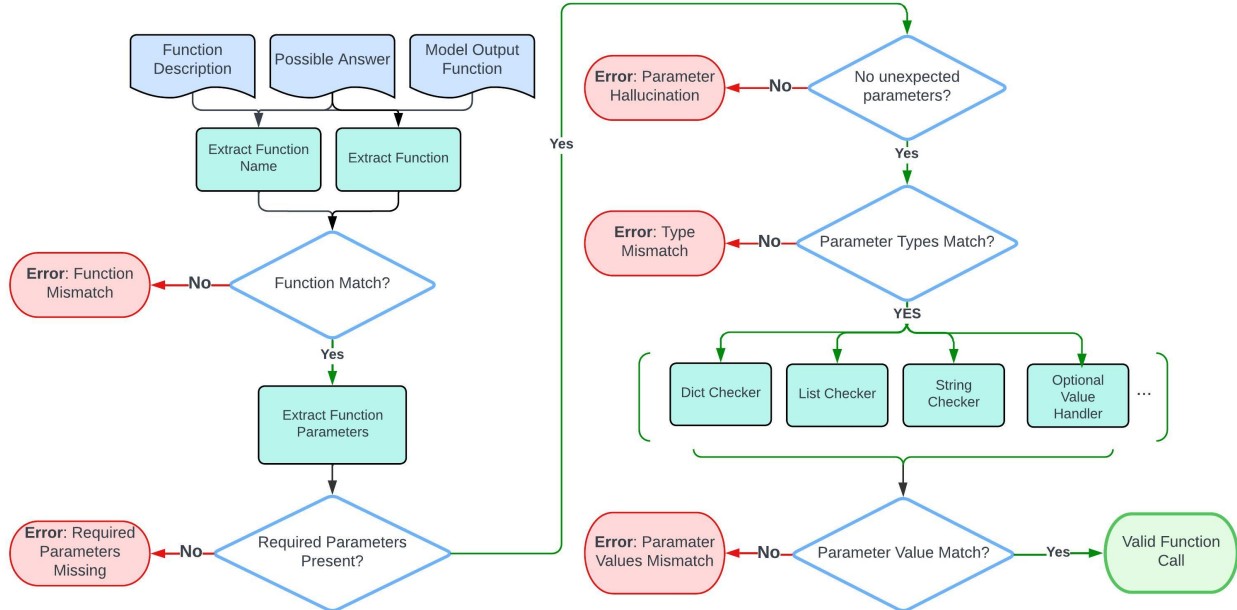

Figure 9: BFCL AST Substring Matching procedure. The model response is parsed to extract function information in the systematic manner outlined above.

**Parameter Values**

INTEGER VS. FLOAT

- **Python only**: an `int` may be supplied where a `float` is expected (Python auto-converts).

- **Java & JavaScript**: when documentation specifies a `float`, the model must output a literal float (e.g. `5.0`); an `int` such as `5` is incorrect.

- Supplying a `float` for an `int` parameter is invalid in *all* languages.

LIST AND TUPLE

- Order matters: `[1,2,3]` $\neq$ `[2,3,1]`. For order-agnostic questions, all permutations of the correct answer are enumerated.

- Type matching is recursive for nested structures; outer and inner element types must satisfy the specification.

STRING

- Comparison is case-insensitive.

- All strings are standardized before checking:
  - whitespace removed,
  - punctuation `,./-_*^` (note: `_` and `^`) stripped.

- **Examples**
  Possible dates: `["20th June", "2023-06-20", "06/20/2023", "Jun.20, 2023"]`
  Possible locations: `["New York City", "NYC"]`

DICTIONARY (`DICT`)

- Key presence and value correctness are checked.

- Key order is ignored (dictionaries are inherently unordered).

- The *list* order of dictionaries matters.

- Within each dictionary, key order does *not* matter.

**Cross-Language Notes**

For Java and JavaScript, strings representing code constructs are converted to Python equivalents using Tree-Sitters before evaluation. During conversion, parameter types are also validated (e.g. a Java `long` must end with `L`).

## I. Function Parameters Distribution

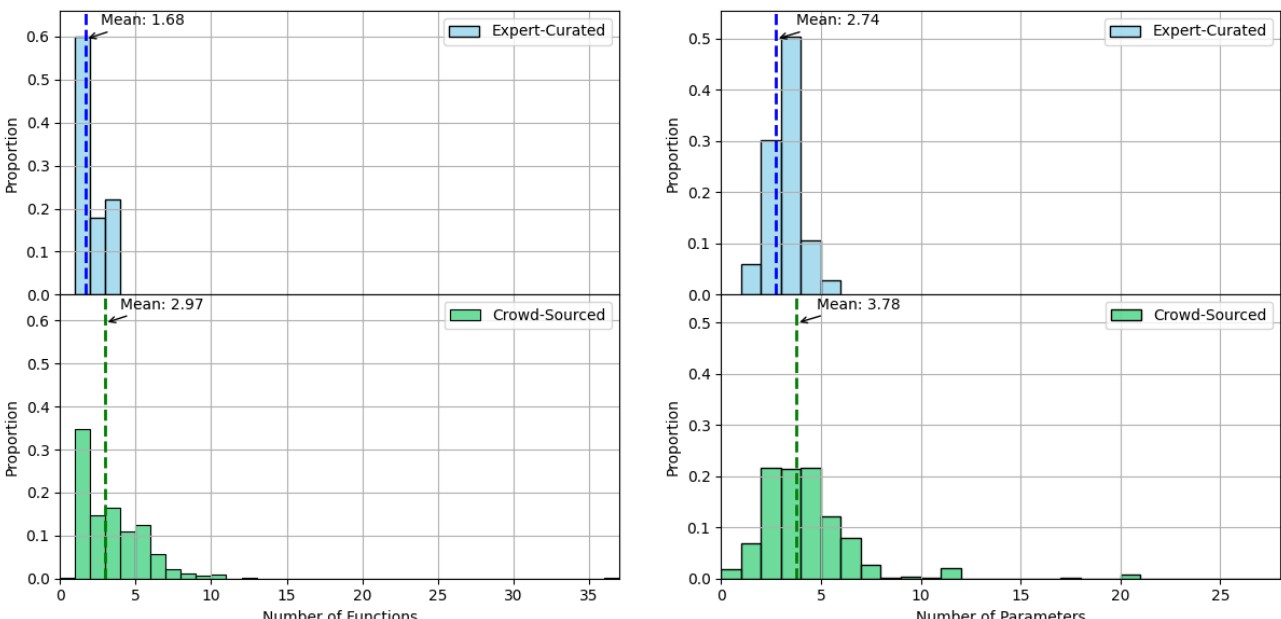

Figure 10: Distribution of functions (left) and function parameters (right) in BFCL dataset between single-turn and crowd-sourced categories. The histograms reveal that the crowd-sourced data entries have broader range and higher mean in both functions and parameters count compared to single-turn scenarios. It's worth noting that in crowd-sourced, we have entries with 37 functions and functions with 21 parameters, where that max number for single-turn is only 3.36 and 3.69, respectively.

## J. Format Instruction Prompt for Agentic Task

This is the additional system prompt that models would receive on agentic dataset entries:

```
"""
For your final answer to the user, you must respond in this format: {'answer': A
    short and precise answer to the question, 'context': A brief explanation of
    how you arrived at this answer or why it is correct}. If you do not know the
    answer, respond with {'answer': 'I do not know', 'context': 'I do not know'}.
     If you think the question cannot be properly answered, response with {'
    answer': 'I cannot answer this question', 'context': A short reason
    explaining why this question cannot be answered}.
"""
```

