# OpenReview forum: "The Berkeley Function Calling Leaderboard (BFCL): From Tool Use to Agentic Evaluation of Large Language Models"
_ICML.cc/2025/Conference — ICML 2025 poster_

### Official Review · Reviewer_krTJ · 2025-03-09

**Overall Recommendation:** 3

**Summary:**

This work introduces function-calling benchmark, a wide-coverage benchmark that evaluates LMs' ability of invoking correct function calls. Multiple parts and types of function calling are included: single-turn, crowd-sourced, multi-turn and agentic tasks. Furthemore, an evaluation method based on Abstract Syntax Tree (AST) substring matching is introduced to provide an appropriate way to evaluate LMs' performance.


Update after rebuttal:
Details as discussed in the authors' rebuttal should be added to later versions of this work. I will maintain my original score.

**Claims And Evidence:**

Yes

**Essential References Not Discussed:**

N/A

**Experimental Designs Or Analyses:**

Yes.

**Methods And Evaluation Criteria:**

The benchmark and evaluation methods make sense.

**Other Comments Or Suggestions:**

N/A

**Other Strengths And Weaknesses:**

Strengths:
- This work introduces a nice benchmark to evaluate LM's function-calling abilities.

Weaknesses:
- More discussions on the comparisons and differences to existing benchmarks (such as Gorilla) should be included.
- There is a lack of clear descriptions of how different types of data instances look like. Figure 2 only seems to show some examples of single-turn ones, while it will be more informative to provide more examples and comparisons of different parts.
- Some parts of the presentation of this work can be further improved, for examples, some tables (Table 1) and figures (Figure 1) are difficult to read.

**Questions For Authors:**

N/A

**Relation To Broader Scientific Literature:**

It provides a reasonable benchmark to evaluate LM's function calling abilities.

**Theoretical Claims:**

N/A

---

> ### Author Rebuttal · Authors · 2025-04-01
>
> We thank the reviewer krTJ for the time to review our paper and the comments. We emphasize the motivation, purpose, and contribution of our work. We address your questions as follows.
>
>
> 1. **Need more discussions on the comparisons and differences to existing benchmarks**
>
> A: Although in recent times quite a few benchmarks have been proposed to test LLM’s ability to perform function calling, in this work, we focus on those designed to evaluate a model’s native function-calling capabilities, rather than methods that rely on prompt-based or code-generation approaches (e.g., Nexus Raven uses a prompt-based protocol, and AppWorld emphasis code generation capability). Among benchmarks that do evaluate native function calling, many such as App Blend and API Bench focus solely on single-turn interactions. Although TinyAgent addresses nested function calls, it does so by using placeholder variables instead of letting the model see the actual execution output of earlier calls; in effect, it still operates under a single-turn framework.
>
> For benchmarks that truly cover multi-turn interactions, most are constrained by a narrow domain scope or a limited set of functions. TauBench, for example, supports only 28 functions spanning two domains (airline and retail), and RestBench offers scenarios solely within the TMDB and Spotify domains. The narrow coverage (<150 entries) is more prone to overfitting and does not sufficiently reflect the breadth of real-world function-calling scenarios.
> Lastly, benchmarks like ToolSandBox and TauBench rely on LLMs to simulate user queries. Despite careful attempts to control the user simulator’s behaviour, LLM-based users remain prone to hallucination and instruction-following errors, which can confound evaluation results.
>
> Other works, such as ToolBench, depend solely on Rapid APIs that are subject to high variance in performance, making reproducibility a challenge. While the subsequent StableToolBench version mitigates this by caching or simulating API responses, it continues to rely on LLM-based evaluators for determining response solvability, thus risking model-induced biases and undermining objectivity (e.g., GPT-family models tending to favour responses from their own model). Similar issues exist in T-Eval, which also depends on LLM-based evaluation.
>
> Lastly, current benchmarks uniformly employ LLM-curated user queries, limiting their ability to accurately reflect genuine user interactions.
>
> Our benchmark directly addresses these shortcomings by incorporating deterministic evaluation metrics, an extensive and diverse set of robust multi-turn interactions, and a unique multilingual dataset derived from real-world, user-contributed queries that have more than 15 languages represented in user queries, including Chinese, French, Japanese, and Korean, etc. Additionally, on top of Python and REST (which are commonly covered in existing benchmarks), we also have entries in Java and JavaScript for more diverse programming languages.
>
> Collectively, these enhancements enable a more comprehensive, fair, and reproducible assessment of LLM function calling, setting our benchmark apart from existing efforts.
>
> You can find all the datasets and the codebase to run them here: https://anonymous.4open.science/r/FCL. Our evaluation scripts support almost all popular models (100+ proprietary models, and 85+ open-source models).
>
> 2. **Need more descriptions and examples of how different types of data instances look like, especially on multi_turn**
>
> A: A set of example test entries and corresponding ground truth outputs for each test category is available here: https://anonymous.4open.science/r/FCL/TEST_EXAMPLES.md

---

### Official Review · Reviewer_dutu · 2025-03-13

**Overall Recommendation:** 2

**Summary:**

* The paper proposes a benchmark for the task of function calling ("FC") i.e., given a prompt and a set of available functions, to predict the correct sequence of function calls along with necessary parameters that accomplishes the prompt requirement.
* The benchmark evaluates FC ability along various categories: single-turn vs. multi-turn, relevance, various programming languages (e.g., python, java, REST), etc.
* An important aspect of the paper is the proposed evaluation methodology. Here, the paper introduce various hand-crafted methods to evaluate depending on the scenario (e.g., primarily AST for "simple" calls, state- and response-based evaluation for multi-turn calls)
* Finally, the paper evaluates existing LLMs (both with/without native support for FC) and presents some interesting analysis e.g., performance across categories, breakdown of errors in various categories.

**Claims And Evidence:**

The claims are generally quite clear. The only issue I have is multiple claims that downplay complexity of existing benchmarks. For instance :
* "other benchmarks fail to capture ... parallel invocations" (L17): Many recent benchmarks appear to consider more complex invocations compared to the proposed leader e.g., ToolBenchv2, TinyAgent consider invocations as a directed graph
* "failure to evaluate abstention" (L105): Once again, many benchmarks do consider this e.g., ToolBench has >16K APIs, majority of which are irrelevant

**Essential References Not Discussed:**

Essential references are discussed. However, another major concern: a more elaborate comparison and discussion of the proposed leaderboard compared to dozens of other FC leaderboards. For instance, see Table 1 in "Tool Learning with Large Language Models: A Survey", Qu et al., 2024. I believe the authors in the related work discussion (Sec. 2), only present a small subset.

**Experimental Designs Or Analyses:**

This is generally clear and I appreciate the paper's breakdown/analysis of >10 models on the proposed leaderboard. I have a major concern here:
* Did the authors run the evaluation of the models in Table 1, or was is run by the developers of the models? Because, the proposed FCL data has been publicly available for some time now (esp. simple, multiple, ...). I'm concerned of train/test contamination i.e. the evaluated models were trained on parts/entirety of the proposed data. To be honest, this is challenging to verify, since numerous models evaluated (esp. the top-performers) are proprietary and do not make available details on the training methodology. Nonetheless, if this is possible, I fear the analysis of the paper might lead to conclusions on overfit data.

**Methods And Evaluation Criteria:**

The paper does not propose a method per se. Rather, the emphasis of methodological contributions would be the hand-crafted rules to verify. One bit I found somewhat unclear:
* AST substring matching (Sec. 4.1): Although this section appears clear for the case of Python calls in "simple" category, it's unclear beyond. Specifically, if we have predicted sequence $A = [a_1, a_2, ..., a_m]$ and GT sequence $B = [b_1, b_2, ..., b_n]$, it's unclear how the association is done.

**Other Comments Or Suggestions:**

Nitpicks on writing:
* L261: mentions please refer to Section 5.1, within Section 5.1.
* L252 "from some other people's blog posts": cite please

**Other Strengths And Weaknesses:**

No other comments here.

**Questions For Authors:**

Important questions that influence my score (repeating from above):
* Since the evaluated models (e.g., gpt-4o) are proprietary, and the dataset has been publicly available for some time now, how can we ascertain that there was no overlap between the models' training data and the proposed dataset?
* Given that there are >10 FC benchmarks (see Table 1 in "Tool Learning with Large Language Models: A Survey", Qu et al., 2024), what is the novelty introduced by this benchmark? After all, there are already benchmarks that are multi-turn (e.g., AppWorld), require real-world APIs (e.g., ToolBench), require parallel invocations (e.g., TinyAgent).

**Relation To Broader Scientific Literature:**

The main contribution of the paper is the diversity of FC scenarios considered. For instance, compared to some related benchmarks:
* ToolBench: considers complex FC calls, but restricted to REST APIs (here, it considers python, java, REST)
* TinyAgent: also considers complex FC calls, but restricted to single-turn pythonic calls
* AppWorld: requires long and stateful multi-turn interactions, but it's within a sandbox environment

**Theoretical Claims:**

No theoretical claims in the paper.

---

> ### Author Rebuttal · Authors · 2025-04-01
>
> We thank the reviewer dutu for the time to review our paper and the comments. We address your questions as follows.
>
> 1. **Need More Discussions on the Comparisons and Differences to Existing Benchmarks**
>
> A: Please see our rebuttal comment to reviewer krTJ.
>
> 2. **Concern Over Potential Train/Test Data Contamination**
>
> A:
>
> The scores reported are all run by the authors. We make sure all the scores are reproducible, and publicly archive all the results & scores we obtained, which the community finds valuable.
>
> As the reviewer noted, the Single Turn dataset has indeed been publicly available for some time, making it difficult to prevent developers from potentially training on these test sets. Nonetheless, our Crowd-Sourced dataset was designed not only to capture more authentic user interactions but also to detect potential data contamination. By updating the Crowd-Sourced dataset on a monthly or quarterly basis, we make it considerably harder for model providers to anticipate and train on the test set. In principle, if a model has not been trained on our public test sets, it should exhibit comparable relative performance across both the Single Turn dataset and successive checkpoints of our Crowd-Sourced dataset.
>
> Lastly, in the case of open-source models, we can calculate perplexity to identify signs of data contamination (see Section 5.5 for more details).
>
> 3. **Unclear AST Substring Matching Procedure for Predicted vs. Ground Truth Sequences**
>
> A:
>
> By definition, parallel tool calls are considered to execute simultaneously and, therefore, the order of calls does not matter. If order does matter (e.g., strictly sequential calls), the model must emit one function call at a time and await completion before issuing the next. Consequently, for a predicted sequence A = [a_1, a_2, ..., a_m] and a ground-truth sequence B = [b_1, b_2, ..., b_n], we do not require strict positional alignment. Instead, any predicted call a_i can be matched with any ground-truth call b_j. Our evaluation then adopts an “all-or-nothing” criterion: if at least one ground-truth call is not matched by any predicted call, the entire evaluation is considered a failure. This ensures the model correctly identifies all necessary calls, regardless of order.
>
> On the parameter values,
> For integer, float:
> - For Python tests only, we allow the use of `int` values for Python parameters expecting `float` values to accommodate the Python auto-conversion feature from `int` to `float`.
> - For non-Python languages (Java and JavaScript), if the function documentation specifies `float` parameter, then it should be a `float` in the model output (such as 5.0); an `int` (such as 5) will not be correct.
> - Supplying `float` value for `int` parameter is not allowed in any language.
>
> For List, Tuple:
> - Order matters and the elements must match exactly. For example, `[1,2,3]` is not equal to `[2,3,1]`. So for questions where the order in the list doesn't matter, permutations of the possible answer are used to accommodate this situation.
> - Note that the type match extends recursively for nested data structures (List or Tuple), where both the outer type and the inner types of elements must match the specified requirements.
>
> For String:
> - The evaluation process is case-insensitive.
> - All strings will be standardized before checking. This applies to both the model output and the possible answers.
>   - All white space is removed.
>   - A subset of punctuations `,./-_*^` are removed to make the evaluation more robust and accurate.
> - For example,
> - Possible Date `["20th June", "2023-06-20", "06/20/2023", "Jun.20, 2023"]`
> - Possible Location `["New York City", "NYC"]`
>
> For Dict:
> - The evaluation focuses on the key presence and the accuracy of associated values as per the possible answers.
> - Ordering within dictionaries is not considered due to they are inherently unordered.
>
> For lists of dictionaries:
> - While the ordering of dictionaries is considered (since it's a List), the order of key-value pairs within each dictionary is not.
>
> This method works for other languages as well. For Java and JavaScript, the evaluation procedure uses tree sitters to convert the string version of a Java/JS type into their corresponding Python format, since we perform checking with Python data types. For example, a `HashMap` in Java will be converted to a `dict` in Python. During this process, the converter will also perform the type checking for those parameters; e.g. if the parameter should be a `long` type in Java, the converter will check to make sure that the string input does have an `L` at the end (because otherwise, it wouldn't be a valid Java `long` type).

---

### Official Review · Reviewer_VP8g · 2025-03-13

**Overall Recommendation:** 2

**Summary:**

This paper describes a benchmark for evaluating function calling capabilities of large language models. It includes different function calling settings such as single-turn, multi-turn, multiple functions in a single turn, parallel functions etc. The evaluation setup defines execution and non-execution based metrics and includes performance comparison of several open weight and closed weight models on the different function calling settings.

## Update after rebuttal:
The FCL benchmark is valuable for the function calling research community. However, the information about so many tasks and their details is scattered in multiple git repos and blogs. This paper would benefit by being a single source of all the information that exists in v3+agentic, so while I appreciate the extra pointers in the rebuttal, I would encourage adding those to the paper. I will keep my score with the current form of writing.

**Claims And Evidence:**

This is a benchmark paper, so most claims have been made based on the evaluation on the dataset.

**Essential References Not Discussed:**

A detailed comparison with existing benchmarks such as StableToolBench, ToolSandbox and many others covered [here](https://github.com/quchangle1/LLM-Tool-Survey) would be very helpful to understand the novelty of this benchmark. The introduction as well as related work section mentions some but is missing many. For example, StableToolBench does capture real-world use cases and has a good number of data samples with single as well as multiple API invocations.

**Experimental Designs Or Analyses:**

The experimental design for prompting the LLMs to generate function calls and evaluating those on different function call setups described is as expected for this task.

**Methods And Evaluation Criteria:**

The propose a set of evaluation metrics such as AST based accuracy, state-based comparison and they make sense.

**Other Comments Or Suggestions:**

1. Minor typo: Line 260 - For an explanation of the (FC) and (Prompt) annotations in the model names, please refer to Section 5.1. : This is in section 5.1.
2. The paper ends abruptly.

**Other Strengths And Weaknesses:**

1. While the benchmark is useful and the experiments section is comprehensive, it would be good to understand the detailed differences/similarities with existing benchmarks.
2. Detailed statistics of the dataset are missing. For example, for the 200 multi-turn long context examples, what is the average number of turns, or what is the maximum context length?
3. Information about the API domains is missing. For example, appendix E.3 says the multi-turn dataset "spanned eight domains, such as Vehicle Control, Trading Bots, Travel Booking, and the File System." - what are the other 4?
4. Details of data generation are missing. For example, appendix E.3 says "Task Generation: We prompt the model" - which model is this?
5. Details of data validation are missing. For example, is the data validation for multi-turn dataset manual/programmatic/LLM-based?
6. Data sources information is missing. For example, what is the source of the SQL dataset? There are many text2SQL datasets available, is it one of them? What is the source of the database?
7. No information on where to find this dataset is mentioned.

**Questions For Authors:**

1. What makes the Agentic dataset different from others? They are all function calls after all?
2. Memory dataset: What is memory here? external database? So in an agentic setting, we expect the end user to be aware of the memory component and ask for using it for some utterances? Can you give some examples of this dataset?
3. For Execution Response Matching, how do you take care of necessary imports etc.?
4. One of the arguments for Response-Based Evaluation is that the state based evaluation does not work for read-only requests. Why? If the state for read-only calls can include the output of the call, state based evaluation can be applicable?
Does it matter that the model generated the result using intrinsic knowledge vs tool calls as long as it is correct?
5. for the case "Failed to Understand Environment State" - how is the environment state provided to the model?

**Relation To Broader Scientific Literature:**

Function calling capability of large language models is an important area of research. There are existing benchmark datasets for this task such as [ToolBench](https://paperswithcode.com/dataset/toolbench) and [BFCL](https://gorilla.cs.berkeley.edu/blogs/13_bfcl_v3_multi_turn.html). The proposed benchmark seems to include additional categories of function calling settings, but most of the proposed settings are similar to those from BFCL.

**Theoretical Claims:**

NA

---

> ### Author Rebuttal · Authors · 2025-04-01
>
> We thank the reviewer VP8g for the time to review our paper and the comments. We address your questions as follows.
>
> 1. **The Proposed Settings are Too Similar to Those From BFCL**
>
> A: In the interest of double-blind reviewing, we refrain from commenting on this.
>
> 2. **Need More Discussions on the Comparisons and Differences to Existing Benchmarks**
>
> A: Please see our rebuttal comment to reviewer krTJ.
>
> 3. **Insufficient Dataset Statistics and Details**
>
> A:
> We utilized GPT-4o-0806 throughout the data generation process and incorporated a hybrid validation strategy—combining manual, programmatic, and LLM-based methods—at various stages of data curation. For the FCL benchmark data curation across both single-turn and multi-turn categories, we provide a more detailed description of the methodology here https://anonymous.4open.science/r/FCL/DATA_CURATION_METHOD.md. Additionally, we present comprehensive multi-turn dataset statistics and visualizations https://anonymous.4open.science/r/FCL/DATASET_STATISTICS.md, including distributions of turn counts, input token usage, and other key characteristics across multi-turn categories.
>
> 4. **Where to find the dataset**
>
> A: You can find all the datasets and the codebase here: https://anonymous.4open.science/r/FCL
>
> 5. **Clarifying What Makes the Agentic Dataset Unique**
>
> A:
> Apart from the statistics that Agent Dataset has a much higher number of turns needed and that more write (as opposed to read) operations are involved, the Agentic Dataset uniquely evaluates tools frequently integrated with LLMs by specifically focusing on core agentic capabilities, independent of their narrow functional uses. Unlike general-purpose function calling evaluations, which typically assess context-free functions (e.g., a 'pause_music' command relevant primarily to virtual assistants like Siri or Alexa), the Agentic Dataset targets fundamental, broadly applicable agentic skills such as web browsing, memory management, and database querying. Take memory management as an example, the fact that it spans across multiple sessions is something you would never see in a normal/standard multi-turn dataset, yet it is critical for true agentic behaviour.
>
> These abilities constitute the backbone of effective LLM agents but have not yet been systematically examined at scale. While there are attempts to evaluate efficient incorporation of web search knowledge or text-to-SQL, our work is unique in systematically treating these functionalities as integral tool calls within the broader agentic paradigm rather than standalone tasks.
>
> 6. **Defining “Memory” and Its Role in the Memory Dataset**
>
> A: Memory refers to how the model retains and utilizes information from previous interactions so that it can provide coherent, contextually relevant responses over multiple sessions. In practical deployments (e.g., chatbots), relevant conversation data will be stored in an external database. For example, a healthcare agent might store a patient’s reported allergies or lab results for reference in future conversations. This stored information is generally not visible to the user, but they will be given a “wipe memory” command.
>
> The Memory Dataset in our work is designed to test the model’s ability to:
> - Identify which pieces of information from past interactions need to be stored.
> - How to find the right balance between multiple contexts when the memory database is limited in size or entry count.
> - Retrieve those details as necessary to respond to new queries in a later session.
>
> 8. **Evaluating Read-Only Calls: Response-Based vs. State-Based Approaches**
>
> A:
> State-based evaluation is limited for read-only requests because these requests do not fundamentally alter the environment. For instance, if a user wants to list files in a directory, using commands like `cd` and `ls` repeatedly does not change the underlying filesystem state in a way that would distinguish one action from another, so a traditional state-based metric offers little insight.
>
> Moreover, incorporating outputs of read-only calls directly into the state essentially converts the evaluation into a response-based evaluation.
>
> We want to measure whether the model invokes the right tools at the right times, instead of relying on outdated intrinsic knowledge. By focusing on response-based evaluation for these scenarios, we hope to more accurately gauge the model’s real-world tool usage and ensure it is actually calling necessary functions rather than guessing or leveraging intrinsic information.
>
> 9. **How the Model Receives and Understands Environment State**
>
> A: The model won’t be directly informed of the entire state of the environment. Instead, we equip the model with functions that can query partial or entire states of the environment, such as ‘ls’ in a file system or ‘display_tire_pressure' in a vehicle control system. We want the model to not take the state information for granted, but instead actively explore the state and retrieve as needed.

---

> > ### Comment · Reviewer_VP8g · 2025-04-03
> >
> > The FCL benchmark is valuable for the function calling research community. However, the information about so many tasks and their details is scattered in multiple git repos and blogs. This paper would benefit by being a single source of all the information that exists in v3+agentic, so while I appreciate the extra pointers in the rebuttal, I would encourage adding those to the paper.

---

> > > ### Author Response · Authors · 2025-04-06
> > >
> > > Thank you for acknowledging the value of the FCL benchmark. We agree that it's not ideal that the details for each task involved are currently scattered across multiple public blog posts (v1, v2, and v3), and we recognize that it can be cumbersome for readers to navigate multiple sources. We will absolutely address this by consolidating all relevant information into a centralized section in the paper. However, due to the constraints of the camera-ready process, we are unable to make these updates at this moment. We appreciate your feedback and will incorporate these improvements in the final version.

---

### Official Review · Reviewer_fXJo · 2025-03-14

**Overall Recommendation:** 2

**Summary:**

This paper proposes a new benchmark FCL for LLM’s function calling, which contains a ‘single-turn’ dataset, a ‘crowd-sourced’ dataset, a ‘multi-turn’ dataset, and an ‘agentic’ dataset.

**Claims And Evidence:**

The authors claim that "Despite the importance of function calling, there isn’t a standard benchmark to evaluate function calling abilities in a wide range of real-world settings."

I believe that the authors make an unfair comparison in Section 2, Related Work. For example, while Appworld is a simulated API, it provides high-quality human-annotated executable environments and test cases, ensuring fairness in execution and enabling reinforcement learning training. Additionally, ToolBench includes 126,486 instances and 16,464 real-world APIs that can be used to evaluate zero-shot tool calling abilities.

**Essential References Not Discussed:**

ToolBench and StableToolBench are large-scale datasets for tool learning using real APIs. These should be fairly compared and discussed in relation to the contributions of this paper.

**Experimental Designs Or Analyses:**

From Table 1, it is evident that the score distribution between different dataset subsets is very uneven. Many models in the Single Turn, Crowd Sourced, and Hallucination Measure categories score close to 100, which raises doubts about whether these scores can effectively differentiate model capabilities (especially given the rapid progress in the LLM field). In contrast, the best model under the Agentic category for the memory column scores only 12 points, which calls into question the benchmark's validity.

**Methods And Evaluation Criteria:**

I believe that the evaluation methods proposed in this paper are generally reasonable and can reflect the ability of LLMs to call tools.

**Other Comments Or Suggestions:**

This paper should include a conclusion section. The asterisks in lines 238-239 seem to be a markdown structure output from LLMs and should be removed.

**Other Strengths And Weaknesses:**

I believe this paper does not have significant advantages over existing benchmarks, and the insights derived from its evaluations appear to be obvious (e.g., GPT-4o is stronger than GPT-4o-mini).

**Questions For Authors:**

See above.

**Relation To Broader Scientific Literature:**

This paper introduces a new benchmark and a new evaluation metric using AST, which could be beneficial to the development of the community.

**Theoretical Claims:**

None.

---

> ### Author Rebuttal · Authors · 2025-04-01
>
> We thank the reviewer for the time to review our paper and the comments. We emphasize the motivation, purpose, and contribution of our work.
>
> 1. **Need more discussions on the comparisons and differences to existing benchmarks**
>
> A: Please see our rebuttal comment to reviewer krTJ.
>
> 2. **Uneven score distribution across different dataset subsets. Near-perfect scores for single turn while extremely low score for agent memory category**
>
> A: Single-turn tasks remain critical for two reasons - first, they form the foundational building blocks for more complex, multi-turn, and agentic workflows. Second, they help weed out the “weaker” models.
> Despite rapid progress in the LLM field, our updated results show that smaller models (e.g., 27B, 7B, 3B, 1B) still struggle on single-turn entries, indicating the ongoing need for these categories to differentiate model capabilities and guide their development. Even “flagship” models occasionally fail on certain single-turn tasks (e.g., claude-3.7 can’t do any parallel tool calls, and  Llama-3.3-70B cannot refrain from tool call when the query is asking for weather information while the only function available is get_coordinate_by_city), demonstrating that near-perfect scores in one subset do not guarantee universal mastery.
>
> As for the performance gap within the Multi Turn or Agentic category, these tasks are deliberately much more challenging, requiring context retention, dynamic planning, and advanced reasoning skills. Our findings reflect that while many models excel at basic single-turn interactions, they still struggle with multi-turn scenarios. The best model in multi_turn_base can only get 60% correct, and even lower on the more advanced miss_func or miss_param settings, and only 12% in the agentic memory subcategory.
>
> We believe these results underscore the value of our benchmark: it captures the full spectrum from fundamental single-turn tasks to complex agentic use cases. To further illustrate this progression, we have updated [Table 1](https://anonymous.4open.science/r/FCL/Table_1.jpg) with more model sizes. And Figure 7 in the appendix shows how performance on single-turn tasks has steadily improved across multiple checkpoints for flagship models. Together, these additions demonstrate the ongoing relevance and evolving nature of our benchmark across different model scales and capabilities.
>
> 3. **Insights derived seem obvious**
>
> A: The insight is not just the relative position of models (e.g., 4o better than 4o-mini), but rather to unearth interesting patterns, such as the fact that Claude-3.7 fails to handle parallel tool usage… These kinds of fine-grained observations have proven to be valuable to developers of large language models. Indeed, leading LLM providers have cited this work to identify and focus their post-training efforts based on the observations of this benchmark. For example, Google refined Gemini’s parallel tool call mechanism based on our findings, while Meta enhanced Llama’s handling of precise parameter value types by analyzing traces from our dataset. This evidence demonstrates that our work generates concrete, practical insights, rather than just confirming that larger models outperform smaller ones.

---

### Decision · Program_Chairs · 2025-05-01

**Decision:**

Accept (poster)

**Comment:**

This paper is about Berkeley Function Calling Leaderboard, a well known and popular benchmark for evaluating the function-calling capabilities of Large Language Models. Given this is a blind review, it was confusing to few reviewers on how this work compares to well known BFCL leaderboard. All other concerns/questions from reviewers are addressed to a good extent in the authors rebuttal. Therefore, I recommend accepting this work.